# Field calibration of electrochemical $NO_2$ sensors in a citizen science context

Bas Mijling[1], Qijun Jiang[2], Dave de Jonge[3], Stefano Bocconi[4]

[1]Royal Netherlands Meteorological Institute (KNMI), Postbus 201, 3730 AE, De Bilt, The Netherlands
[2]Laboratory of Geo-Information Science and Remote Sensing, Wageningen University & Research, Droevendaalsesteeg 3, 6708 PB Wageningen, The Netherlands
[3]Public Health Service of Amsterdam (GGD), Nieuwe Achtergracht 100, 1018 WT, Amsterdam, The Netherlands
[4]Waag Society, Nieuwmarkt 4, 1012 CR, Amsterdam, The Netherlands

*Correspondence to*: Bas Mijling (mijling@knmi.nl)

**Abstract.** In many urban areas the population is exposed to elevated levels of air pollution. However, real-time air quality is usually only measured at few locations. These measurements provide a general picture of the state of the air, but they are unable to monitor local differences. New low-cost sensor technology is available for several years now, and has the potential to extend the official monitoring network significantly even though the current generation of sensors suffer from various technical issues.

Citizen science experiments based on these sensors must be designed carefully to avoid generation of data which is of poor or even useless quality. This study explores the added value of the 2016 Urban AirQ campaign, which focused on measuring nitrogen dioxide ($NO_2$) in Amsterdam, the Netherlands. 16 low-cost air quality sensor devices were built and distributed among volunteers living close to roads with high traffic volume for a two-month measurement period.

Each electrochemical sensor was calibrated in-field next to an air monitoring station during an 8-day period, resulting in $R^2$
ranging from 0.3 to 0.7. When temperature and relative humidity are included in a multilinear regression approach, the $NO_2$ accuracy is improved significantly, with $R^2$ ranging from 0.6 to 0.9. Recalibration after the campaign is crucial, as all sensors show a significant signal drift in the two-month measurement period. The measurement series between the calibration periods can be corrected in hindsight by taking a weighted average of the calibration coefficients.

Validation against an independent air monitoring station shows good agreement. Using our approach, the standard deviation
of a typical sensor device for $NO_2$ measurements was found to be 7 μg m$^{-3}$, provided that temperatures are below 30°C. Stronger ozone titration at street sides causes an underestimation of $NO_2$ concentrations, which 75% of the time is less than 2.3 μg m$^{-3}$.

Our findings show that citizen science campaigns using low-cost sensors based on the current generations of electrochemical $NO_2$ sensors may provide useful complementary data on local air quality in an urban setting, provided that experiments are
30 properly set up and the data are carefully analysed.

# 1 Introduction

Because air pollution is difficult to measure, instrumental and operational costs of official measurement stations are usually high. Air quality networks in cities, if present at all, are therefore usually sparse. Diffusive sampling is a common addition to these real-time measurements and are successfully used to monitor local differences (see e.g. Cape, 2009). However, these differences are poorly attributed to an emission source due to the long averaging time of these measurements (usually 4-weekly). Emerging low-cost sensor technology has the potential to extend the official monitoring network significantly, and improve our understanding of local urban air pollution. Miniaturized and affordable sensors potentially enable citizens to measure their environment in more detail in space and time (Kumar et al., 2015). Most commercially available sensors, however, suffer from various technical issues which limit their applicability. Despite their limitations many experiments are done with air quality devices containing these sensors, often by motivated but not necessarily scientifically trained people. Comprehensive calibration and validation of these devices is crucial (see e.g. Lewis and Edwards, 2016; Lewis et al., 2016), but often overlooked. The resulting poor data quality is of concern to health authorities, scientists and citizens themselves. Several studies have been done to explore the performance of low-cost air quality sensors, e.g. Jiao et al., 2016, Duvall et al., 2016; Mead et al., 2013; Moltchanov et al., 2015. For $NO_2$ monitoring, mostly metal oxide and electrochemical sensors are used (Borrego et al., 2016; Spinelle et al., 2015b; Thompson, 2016). Typical ambient concentrations of $NO_2$ are at part-per-billion (ppb) level. The main problems encountered in $NO_2$ sensor evaluations in these real-world environments are low sensitivity, poor selectivity, low precision and accuracy, and drift. Especially metal oxide sensors are not very stable (Spinelle et al., 2015b; Thompson, 2016) and suffer from lower selectivity. Therefore, in this study, we opted for electrochemical sensors to measure $NO_2$.

Mead et al. (2013) already noted the strong interference of ozone and other ambient factors in electrochemical $NO_2$ sensors. The performance can be increased significantly when adding additional measurements of e.g. temperature and humidity in a regression model or neural network, as shown by e.g. Piedrahita et al. (2014), Spinelle et al. (2015b), Masson et al. (2015). Coping with sensor degradation remains a serious issue. Some studies, such as Jiao et al. (2016), include an additional temporal term in their linear regression which improves the predicted $NO_2$ slightly.

In the following sections we assess the data quality of the 2016 Urban AirQ campaign. As many similar initiatives depending on participating citizens, this campaign was not set up as a strictly controllable scientific experiment such as in the previously mentioned studies. However, we will demonstrate that citizen air quality monitoring using the current generation of electrochemical $NO_2$ sensors may provide useful data of urban air quality, by using a practical method for field calibration and correcting for sensor degradation in hindsight.

# 2 The Urban AirQ project

The Urban AirQ project explores the added value of alternative air quality measurements in the city, by addressing citizens' questions about their local air quality. It focusses on a 2×1 km$^2$ area around Valkenburgerstraat, a primary road in the East-

central part of Amsterdam, see Figure 1. Its dense traffic causes regular exceedances of the European annual limit value for nitrogen dioxide (40 μg m$^{-3}$).

Two town hall meetings were organized in which residents of this area were invited to raise their concerns about air pollution in their neighborhood and to formulate related research questions. Topics included the relation between traffic density and air pollution, the difference between main roads and side streets, the front side of an apartment compared to its backside, the influence of apartment height, and the influence of cut-through traffic at nighttime. The residents were invited to participate in finding answers to their questions by measuring their outdoor air quality with 16 experimental low-cost sensor devices (labeled SD01 to SD16), built for this purpose by Waag Society.

Measurements were done from June to August 2016. Beforehand, the sensor devices were calibrated using side-by-side measurements next to an official air quality measurement station. With a second calibration period after the campaign, individual sensor drift was assessed and compensated in hindsight.

The Urban AirQ experiment is unique in the sense of the used number of devices, the duration of the experiment, the direct involvement of citizens, and the use of open hardware and generation of open data.

**3 Urban AirQ sensor devices**

The concept of the Urban AirQ sensor is building a device with low-cost electronic components which is easy to operate, so citizens can do their own air quality measurements. It builds on the basic design described by Jiang et al. (2016), having an improved power supply, weather resistant housing, WiFi connectivity, and additional sensors for temperature, relative humidity, and particulate matter. The sensor development is part of an open hardware project; detailed technical information can be found at https://github.com/waagsociety/making-sensor.

Central is the microcontroller board (Arduino UNO) which handles the reading of the sensors and sends the data to the WiFi module (ESP8266), see Figure 2.

For NO$_2$ measurements, an electrochemical cell is used from Alphasense Ltd (Essex, United Kingdom). The cell contains four electrodes. The target gas, NO$_2$, diffuses through a membrane where it is chemically reduced at the Working Electrode, generating a current signal. This electric current is balanced by a opposite current from the Counter Electrode. The Reference Electrode sets the operating potential of the Working electrode. The sensor also includes an Auxiliary Electrode, which is used to compensate for baseline changes in the sensor. To get full sensor performance, low noise interface electronics is necessary. An individual sensor board with amperometric circuitry, also provided by Alphasense, is used to guarantee a low noise environment and to optimize the sensor resolution at low ppb levels. The sensor signal is read by a 16-bit analog to digital (A/D) converter (ADS1115). Two sensor devices (SD01 and SD02) contain model NO2-B42F for NO$_2$ measurements, the other 14 contain the newer NO2-B43F sensor.

12 of the 16 sensor devices are also equipped with a Shinyei PPD42NS sensor in order to measure particulate matter optically. The present paper, however, will focus only on the assessment of the $NO_2$ measurements. All devices measure internal temperature and relative humidity (RH) with a DHT22 sensor from Aosong Electronics.

The system is supplied with a 7.5V voltage output adapter and a regulator board which generates 5V for the Arduino and the sensors. The microcontroller consumes a 10 mA current (measured). The PM sensor needs up to 80 mA (measured), the $NO_2$ sensor about 10 mA (measured), and the DHT22 less than 1 mA. The WiFi module peaks periodically to 350 mA when establishing an internet connection.

## 3.1 Averaging and filtering

Raw sensor measurements are stored in a central database on a one minute base. However, the calibration analysis is based on hourly averages to enable direct comparison between the ground truth (also provided as hourly values), and to improve the signal to noise ratio.

The $NO_2$ sensor measurements are done at the Working Electrode ($S_{WE}$) and the Auxiliary Electrode ($S_{AE}$). They are provided as counts from the A/D converter. Sensor readings of temperature and RH are converted according to the indication of the manufacturer to degrees Celsius and percentages respectively.

Raw, hourly averaged, sensor data are shown in Figure 3. The spread in temperature and RH displayed in the raw data is partly explained by the sensor-to-sensor variability. By looking at nighttime temperatures (to eliminate the effect of local heating by exposure to direct sunlight) we see that the internal sensor temperatures are 2-5°C higher than ambient temperature. The devices are not actively ventilated, which means that the energy dissipation of the electronics influences their internal temperature. The variable position of the temperature sensors with respect to these heat sources further explain the variance in temperature and relative humidity.

Careful filtering is needed before the data can be further processed. We have applied the following rules:

- Raw, minute-based, $S_{WE}$ and $S_{AE}$ measurements outside a ±10% range of their mean value during the entire measuring period are considered outliers. This filters out 0.33% of all measurements. This criterion was used for its simplicity and effectiveness. Note that, due to the large offset in the raw $S_{WE}$ and $S_{AE}$ signal, realistic $NO_2$ peak values are still detectable as the corresponding sensor response is still within a 10% bandwidth.

- All readings at sensor temperatures above 30°C are discarded to avoid non-linear temperature dependence of the electrochemical $NO_2$ sensor (see Sect. 4.4). This filters out 4.53% of the measurements during the entire period.

- At least 20 valid minute-based measurements are required to calculate a representative hourly mean. This criterion was found to be a good trade-off between noise reduction by averaging and not losing too many hourly measurements.

During the first calibration period, the sensors were measuring 79% of the time on average. After applying the criteria above, this resulted in 70% valid hourly measurements. During the measurement campaign, the sensors produced 79% valid hourly

measurements on average, with the uptime dropping to 50% in places were sensors experienced connectivity problems due to limited range of the participant's WiFi network.

## 3.2 Calibration periods

Calibration of the sensors devices have been done by placing the 16 sensors side by side on the rooftop of the air quality station at Vondelpark, operated by the Public Health Service of Amsterdam (GGD). This station is classified as a city background station. It measures nitrogen dioxide, nitrogen monoxide (NO), ozone ($O_3$), particulate matter ($PM_{10}$, $PM_{2.5}$, particle number and size distribution), black carbon, and carbon monoxide (CO). For NO and $NO_2$ measurements, GGD alternates a Teledyne API 200E and a Thermo Electron 42I $NO/NO_x$ analyser, both based on chemiluminescence. The validated measurements used in this study are considered to be the ground truth. The calibration period spanned several days to be able to test the sensors under a wide range of ambient conditions. To assess the stability of the calibration, the sensors were brought back after the two-month measurement campaign to the calibration facility for a second calibration period. The Urban AirQ campaign consisted therefore of three phases.

The first field calibration period at GGD Vondelpark station started at 2 June 2016, 00h LT (local time), and ended at 10 June 2016, 10h (8.5 days; 204 hours). Due to connectivity problems sensor data were missing between 4 June 19h and 6 June 9h.

During the following citizen campaign, 15 sensors were distributed among the participants. One sensor (SD03) was kept at the Vondelpark station as a reference. The first sensor was installed and connected at 13 June 2016, 18h, and the last sensor connected at 17 June 2016, 17h. At 15 August 2016, 9h, the first sensor was disconnected, and at 16 August 2016, 18h, the last sensor was disconnected. In this 1537-hour period the devices produced 1204 valid hourly measurements on average.

The second field calibration period at GGD Vondelpark station started at 18 August 2016, 15h, and ended at 29 August 2016, 00h (10.4 days; 249 hours). Due to connectivity problems sensor data were missing between 26 August 12h and 27 August 11h.

Figure 4 shows the distribution of temperature, relative humidity, $NO_2$, and $O_3$ during the different periods. Looking at the 75[th] percentile of the distributions, the calibration periods are characterized by higher temperatures and ozone levels than the campaign period. The range of $NO_2$ concentrations at the Vondelpark station in the calibration periods is larger than in the campaign, reaching more frequently higher $NO_2$ values. During the campaign the sensors are closer to the GGD station at Oude Schans, where measured $NO_2$ values are generally a few µg m$^{-3}$ higher than at Vondelpark. The Oude Schans site does not measure ozone.

## 4 $NO_2$ calibration

Electrochemical sensors such as the Alphasense NO2-B series are known to be sensitive to interfering species and ambient factors. Especially ozone, temperature, and relative humidity influence the sensor reading (see e.g. Spinelle et al., 2015a).

### 4.1 Explaining the $NO_2$ sensor signal

To understand better the behavior of the $NO_2$ sensor, we study its sensitivity to different ambient factors. We use the first calibration period to test the correlation of the measured $S_{WE}$ and $S_{AE}$ signal with $NO_2$, ozone, temperature and humidity by making a best fit though the hourly time series, e.g.

$$S_{WE}(t) = c_0 + c_1 NO_2(t) \tag{1}$$

Temperature and RH were not readily available from the GGD Vondelpark station data. We take temperature and RH from the average readings from the DHT22 sensors instead, which better reflect the internal sensor conditions than ambient air measurements.

Figure 5 shows scatter plots for an average performing sensor and the $R^2$, the coefficient of determination. The measured $S_{WE}$ signal can be explained by ambient $NO_2$ ($R^2$=0.20), but better by its anti-correlation with ozone ($R^2$=0.49). Temperature

alone is an even better predictor for the sensor signal ($R^2$=0.73), because of the sensors's direct dependence on temperature, and indirect dependence on temperature (being a reasonable proxy for both $NO_2$ and $O_3$ concentrations). Also the correlation with relative humidity is very strong ($R^2$=0.73). The measured $S_{WE}$ signal can best be explained as a linear combination of $NO_2$, $O_3$, T, and RH together, resulting in a correlation of 0.98 ($R^2$=0.96).

The $S_{AE}$ signal is practically insensitive to $NO_2$. This suggests that a combination of $S_{WE}$ and $S_{AE}$ is more sensitive to $NO_2$

and less to the other interfering factors, as intended by the manufacturer.

### 4.2 $NO_2$ calibration models

For $NO_2$ measurements, the sensor manufacturer suggests to correct both Working Electrode and Auxiliary Electrode for a zero-offset with $S_{WE,0}$ and $S_{AE,0}$ respectively. Then a sensitivity constant $s$ is applied to convert from mV to ppb $NO_2$:

$$NO_2[ppb] = \frac{(S_{WE} - S_{WE,0}) - (S_{AE} - S_{AE,0})}{s} \tag{2}$$

In practice, the factory-supplied constants $S_{WE,0}$, $S_{AE,0}$, and $s$ do not result in realistic values of $NO_2$, see e.g. Cross et al.

(2017). As an alternative, we propose a linear combination of the signals $S_{WE}$ and $S_{AE}$ (calibration model A):

$$NO_2[\mu g\ m^{-3}] = c_0 + c_1 S_{WE} + c_2 S_{AE} \tag{3}$$

The coefficients $c_1$ and $c_2$ are determined with data from the calibration period using ordinary least squares (OLS). As can be seen from the fit results in Table 1, within the batch of sensors there is a large variability of direct sensitivity to ambient $NO_2$. During the calibration period, hourly ozone values (also taken from the Vondelpark station) happened to be a good proxy for the ambient $NO_2$ concentration: $NO_2(t) = 44.6 - 0.40 \cdot O_3(t)$ in [$\mu g\ m^{-3}$], with $R^2$ of 0.49.

When compared with Table 1, it can be seen that direct sensor readings from a fair part of the sensors cannot outperform this result. To improve the results we use additional measurements and their statistical relation to $NO_2$. We fit different

calibration models with multiple linear regression (using OLS). The calibration models which were tested are listed in Table 2.

Temperature and RH are taken from the DHT22 sensor. Note that there is no need to calibrate the individual T and RH sensor signals beforehand; the calibration coefficients for $NO_2$ are determined for the specific set of all sensors in the box. However, this means that if an individual sensor is replaced, new calibration parameters for the sensor box have to be derived.

### 4.3 Calibration results

A complete overview of the regression coefficients and their error estimates for all models can be found in the supplement. The sign of the calibration parameters can be easily understood. As the electrochemical $NO_2$ sensor loses sensitivity at higher temperatures (see the negative slope in Figure 7(b) for temperatures below 30°C), coefficients $c_3$ are positive to compensate for this effect. The additional sensor response due to cross-sensitivity with ozone is compensated by negative values for $c_5$.

From the fit results we see that Model B (including RH) performs better than Model A, but Model C (including T) outperforms Model B. When both RH and T are included (Model D) the results of Model C are marginally improved. This can be understood in terms of a strong sensor dependence on temperature, a weak dependence on RH, and the collinearity between temperature and RH. Note that measuring RH is essential for guarding the data quality of electrochemical sensors, as these sensors are very sensitive to *sudden changes* in RH, see e.g. AAN-110 (2013) and Pang et al. (2016).

The best calibration results (i.e. $R^2$ values closer to 1) are obtained by including ozone (Model E). The ozone values were obtained from the GGD Vondelpark station, as the sensor devices do not measure ozone themselves.

As local ozone measurements were only available during the calibration periods, we used Model D for the Urban AirQ campaign, i.e. generating an $NO_2$ value based on a linear combination of $S_{WE}$, $S_{AE}$, T, and RH. The regression analysis of Model D and correlation with the $NO_2$ ground truth can be found in Table 3.

The two worst performing sensor devices (SD02 and SD01) contain the older NO2-B42F sensor. The newer NO2-B43F model is designed to have higher sensitivity to NO2 and less interference of ozone. The old sensor model has indeed smaller coefficients for $S_{WE}$ and larger correction terms for ozone (see the $c_1$ and $c_5$ coefficients of model E in the Supplement). This, however, can also be related to their longer operating time, as both sensors have been used in previous experiments for more than a year. Again, it can be seen that even within the same batch of sensors there is a significant spread in performance, around a median value for $R^2$ of 0.83. Figure 6 shows the results for the different calibration models for the average performing sensor SD15. The time series in Figure 6(b) shows clearly how the performance of a typical sensor device improves when temperature and humidity are included in the calibration analysis. The adjusted $R^2$, which corrects $R^2$ for the number of explanatory variables, increases from 0.29 to 0.82. Note that $R^2_{adj}$ is only slightly smaller than $R^2$, as the number of observations ($n \approx 150$) is relatively high compared to the number of regression variables ($k=2…5$).

### 4.4 Dependency on temperature

Calibrated data without temperature filter show occasionally strong negative values, see Figure 7 below. These negative peaks coincide with internal sensor temperatures exceeding 30 °C. This behavior can be explained from the dependency of the electrochemical sensor on temperature becoming non-linear, see Figure 7(b): the sensitivity of the $NO_2$ sensor decreases linearly with temperature up to around 30 degrees, while above 40 degrees the sensor gains sensitivity with rising temperatures. In these regimes, the response of the sensor cannot be described well with our multilinear regression approach. As temperatures during the measurement period only rose occasionally above 30 °C, we decided to filter these measurements out.

### 4.5 Startup time

When a sensor device is switched on for service, the electrochemical cell must be stabilized by the potentiostatic circuit which can take a few hours due to the high capacitance of the working electrode (AAN-105, 2009). Furthermore, when the sensor is transported to another environment the sudden change in RH causes an equilibrium distortion with a relaxation time of about 2h (Mueller et al., 2017). The startup-effect is translated by the calibration model as a strong positive $NO_2$ peak, which should be filtered out. From our sensor data we estimate a stabilization time of 4 hours. Note that this startup effect should not be confused with the response time, which is determined to be less than 2 minutes in Mead et al. (2013) and Spinelle et al. (2015a).

### 4.6 Predictivity, sensor drift, and uncertainty estimation

Almost all electrochemical sensors have some degree of drift because of aging and poisoning (Di Carlo et al., 2011; Hierlemann and Gutierrez-Osuna, 2008). This becomes a serious complication when the drift is in the order of the strength of the signal of interest. The idea of keeping sensor SD03 next to the reference station during the whole campaign was to study sensor degradation in more detail. Unfortunately, the sensor was removed temporarily from 10 to 14 July for service, when it was decided to add a PM module to the device. The increased energy dissipation after the modification (the Shinyei PPD42NS module uses a heater resistor to force a convective flow of sampling air) caused an increase of the internal device temperature by 2.5°C on average. This sudden jump in temperature disrupted the reference time series.

Instead, to assess the short-term stability of the calibration model, we use the first 60% of the measurements from the calibration period (2-7 June) to derive the regression coefficients, and predict the $NO_2$ values for the remaining 40% (8-10 June), see Table 4. The average RMSE increases from 6.5 to 7.0 μg m$^3$ when the regression is used for prediction.

We assess the long-term stability of the sensors with a second calibration period after measurement campaign, again at the Vondelpark calibration site. As can be seen from the distribution of the residuals in Figure 8, most sensors drift significantly in the intermediate two-month period. We describe this degradation effect as a bias $b$ between the mean of the hourly estimated $NO_2$ values $\hat{x}_i$ and the mean of the hourly true $NO_2$ $x_i$ during the calibration period:

$$b = \frac{1}{N}\sum_{i=1}^{N}\hat{x}_i - \frac{1}{N}\sum_{i=1}^{N}x_i \tag{4}$$

and the root-mean-square error (RMSE) of the difference between the bias corrected calibrated measurement and the ground truth. The latter is the same as the standard deviation of the residuals (SDR) $\hat{x}_i - x_i$:

$$\text{SDR} = \sqrt{\frac{1}{N}\sum_{i}\left((\hat{x}_i - b) - x_i\right)^2} \tag{5}$$

As can be seen in Table 5, the bias is mostly positive. Note that sensor SD16 and SD01 had a limited uptime in the second period, which makes their bias and RMS calculation not very representative.

The strongest bias after two months is found for SD02 and SD01. Both are of model NO2-B42F and have been used in others experiments for more than one year. These sensors have also the largest RMSE in the first calibration period (see also Table 3), which is another indication of their poor performance. The range in RMSE of the remaining sensors is 4.5 – 7.2 µg m$^{-3}$ for the first period. The bias corrected RMSE increases to 5.3 – 9.3 µg m$^{-3}$ for the second period. The latter is a more conservative yet more realistic estimation of the precision of the NO$_2$ estimates, as they are based on measurements which

were not used for calibration. Based on our results listed in the last columns of Table 4 and 5, we take 7 µg m$^{-3}$ as a typical uncertainty for the estimated NO$_2$ values.

The increase of SDR is also due to a loss of sensitivity over time. The aging of the sensors can be further investigated by recalibrating the devices, i.e. determining the coefficients of regression model D, using the data of the second calibration period (see the Supplemental Material). All calibration coefficients of $S_{\text{WE}}$ (the only component which has direct sensitivity

to NO$_2$) decrease in value, showing that all sensors suffer from sensitivity loss to NO$_2$. This results in lower $R^2$ values, although the performance loss is partly compensated by the other components in the regression. The older Alphasense models NO2-B42F suffer the largest sensitivity loss, which (although the regression tries to compensate with an increased temperature dependence) result in the worst performance loss in terms of $R^2$.

**4.7 Weighted calibration**

Taking 18 µg m$^{-3}$ as a typical NO$_2$ concentration in an urban environment (Figure 4), the sensor drift as listed in Table 5 is a significant error component, even after a two month period. It is impossible to predict the progressing bias for an individual sensor. However, using the second calibration period we can compensate for signal drift in hindsight. If $\hat{x}_1(t)$ represents the estimated NO$_2$ value at time $t$ based on the first calibration period (starting at $t_1$), and $\hat{x}_2(t)$ the estimated NO$_2$ value based on the second calibration period (ending at $t_2$), the we take for intermediate times $t_1 \leq t \leq t_2$ a weighted average of both

calibrations:

$$\hat{x}(t) = \left(1 - f(t)\right)\hat{x}_1(t) + f(t)\hat{x}_2(t) \tag{6}$$

Assuming that the sensor degradation is linear in time we select

$$f(t) = (t - t_1)/(t_2 - t_1) \tag{7}$$

such that $f(t_1)=0$ and $f(t_2)=1$.

## 4.8 Validation against an independent reference station

Citizen science can be unpredictable, and we were fortunate that sensor SD04 was handed over to an Urban AirQ participant living at Korte Koningsstraat (ground floor), which happens to be 120m from another GGD station at Oude Schans (see Figure 1). The Korte Koningsstraat is a side street away from traffic arteries, whereas Oude Schans also classifies as an urban background location. The proximity to a reference station enabled us to perform an independent validation of the sensor measurements, as the calibration of the sensor is based on side-by-side measurements with Vondelpark station, at 3 km distance. As can be seen from Figure 9, the sensor readings agree very well with the official measurements. Using the weighted calibration explained in the previous section, the measurement bias largely disappears (Table 6). The RMSE (5.3 µg m$^{-3}$) is comparable to the RMSE found during the calibration period. The results give confidence that our calibration method remains valid for similar urban locations, and that our assumption of sensor degradation being linear in time is acceptable.

## 5 Discussion

The Alphasense NO2-B4 sensor is used in many low-cost air quality applications for measuring ambient $NO_2$. As all electrochemical $NO_2$ sensors, it is not very selective to the target gas. The sensor response can be explained well by a linear combination of $NO_2$, $O_3$, temperature and relative humidity signals ($R^2 \approx 0.9$).

As a consequence, a linear combination of the Working Electrode and the Auxiliary Electrode alone give poor indication of ambient $NO_2$ concentrations. The accuracy varies greatly between different sensors ($R^2$ between 0.3 and 0.7). For the Urban AirQ campaign, temperature and relative humidity were included in a multilinear regression approach. The results improve significantly with $R^2$ values typically around 0.8. This corresponds well with the findings of Jiao et al. (2016), who find an adjusted $R^2=0.82$ for the best performing electrochemical $NO_2$ sensor in their evaluation, when including T and RH.

Best results are obtained by also including ozone measurements in the calibration model: $R^2$ increases to 0.9. Spinelle et al. (2015b) used a similar regression and found $R^2$ ranging from 0.35 to 0.77 for 4 electrochemical $NO_2$ sensors during a two-week calibration period, but dropping to 0.03—0.08 when applied to a successive 5-month validation period. Low $NO_2$ values at their semi-rural site partly explains this poor performance, but most likely also unaccounted effects such as changing sensor sensitivity and signal drift.

The sensor devices were tested in an Amsterdam urban background in summertime, with $NO_2$ values ranging from 3 µg m$^{-3}$ to 78 µg m$^{-3}$, and median values around 15 µg m$^{-3}$. During the 3-month period most sensors show loss of sensitivity and

significant drift, ranging from -9 to 21 μg m$^{-3}$. After bias correction we found a typical value for the accuracy of the NO$_2$ measurements of 7 μg m$^{-3}$.

This error consists of several components. The reference measurements by the NO/NO$_x$ analysers have an estimated hourly error of 3.65% (certified validation at a 200 μg m$^{-3}$ NO$_2$ concentration), which would contribute to 0.5 μg m$^{-3}$ under typical

conditions. The low-cost DHT22 sensor has a reported error of 0.5 °C for temperature and 2–5% for RH. For a single measurement, this would contribute to a propagated regression error of approximately 1 μg m$^{-3}$ and 0.5 μg m$^{-3}$, respectively. It should be noted, however, that binning minute-based measurements to hourly averages removes large part of the variability, while determining the best fitting regression model for each sensor device removes large part of the remaining systematical biases. The largest part of the error term is therefore introduced by the linear regression model itself, which

does not include all interfering species or meteorological quantities, and is not able to describe non-linear dependencies of its variables. One should therefore be careful to extrapolate the calibration model for conditions different than the calibration period.

The validation results from Section 4.8 show that the calibration holds well for urban locations with similar NO$_2$/O$_3$ ratios. Neglecting O$_3$ as regression parameter, however, will introduce a bias at locations with different NO$_2$/O$_3$ ratios found e.g.

closer to emission sources. To get a better understanding of the possible impact, we compared hourly ozone measurements from the GGD authorities at Van Diemenstraat (VDS, classified as street station) against Nieuwendammerdijk (NDD, classified as urban background station) during June-August 2016. The relation can best be described by [O$_3$]$_{VDS}$ = 0.87 [O$_3$]$_{NDD}$ + 0.85 (with 0.93 correlation), which means that ozone levels at the street station are typically 13% lower, due to titration of O$_3$ with NO. Due to the sensor's cross-sensitivity for ozone, larger values must be subtracted from its signal when

the ozone concentration increases. This explains the negative sign of the ozone coefficient $c_5$ of model E (see Supplement). Calibration with model D will overcorrect (i.e. subtract too much) for locations which have lower ozone concentrations than at the calibration site, resulting in an underestimation of NO$_2$ concentrations. Using typical values $c_5$=-0.3 and [O$_3$]=60 μg/m$^3$ (75[th] percentile of the distribution during the measurement camping, according to Figure 4) we estimate the underestimation of NO$_2$ at street side as 0.3 × 13% × 60 = 2.3 μg/m$^3$.

The found sensor accuracy after weighted calibration is good enough to provide some complementary spatial information on local air quality between reference stations. When looking at the difference between Vondelpark station and Oude Schans station (both classified as city background stations) in the period June-August 2016, 22% of the hourly measurements differ more than 7 μg m$^{-3}$, and 6% of the hourly measurements differ more than 14 μg m$^{-3}$. These differences increase further when considering road side stations. From this perspective, even sensor devices with an accuracy around 7 μg m$^{-3}$ can contribute to

an improved understanding of spatial patterns. However, it must be further investigated if the calibration method used here would provide realistic estimates for peak values (such as the EU hourly limit value, 200 μg m$^{-3}$).

## 6 Conclusions and outlook

In this study, we examined low-cost electrochemical air quality sensors for citizen urban air quality monitoring. In other words, we evaluated an imperfect air quality sensor in an imperfect scientific experiment. In general, we found that low-cost electrochemical sensors have the potential to complement official environmental monitoring data to help answer questions from the public, which usually cannot be fully answered from official data alone. To reach the potential, however, proper measurement set-up, calibration and recalibration, and data analysis should be guaranteed.

The current generation of low-cost $NO_2$ sensors has some serious issues which trouble straightforward application. To make electrochemical $NO_2$ sensor measurements accurate, careful filtering of the raw data is necessary. There is a strong spread in sensor performance, even if the sensors come from the same batch, which make individual calibration essential. A practical calibration method is measuring side-by-side to an air monitoring station. The accuracy of the measurements can be improved by including temperature and humidity measurements from other low-cost sensors in a multilinear regression approach. It is worth noting that more advanced calibration algorithms such as by Cross et al. (2017) and Mueller et al. (2017) could give better results, but this is not the focus of this paper. It is hard to quantify an optimal length of a calibration period without having a proper understanding of the sensor degradation rate beforehand. The measurement period should be at least a few days to capture the sensors behavior under a wide range of pollution levels and meteorological conditions. Very long calibration periods (in the order of months) will cause sensor degradation issues to interfere with the calibration results.

Startup time of sensors is estimated 4 hours. To avoid nonlinear response of the electrochemical sensor at elevated temperatures, we filter out measurements above 30 °C. This is not a serious restriction for applicability in moderate climates such as in the Netherlands, provided that the sensor is protected from direct sunlight. However, for warmer regions or during heat waves this may reduce the data stream considerably, unless the temperature dependencies are better captured by more advanced regression models.

The calibration seems to be location independent, as long as the $NO_2/O_3$ ratio is comparable. Application at a street side is likely to introduce a small positive bias. Calibration coefficients are not constant in time. During the 3-month period most sensors suffer from significant sensitivity loss and drift. The strongest drift and largest uncertainty are found for the older NO2-B42F sensors. It remains unclear if the worse performance is related to the sensor model or the longer usage in field experiments.

The sensor degradation troubles practical applications in operational urban networks. Smart re-calibration programs are essential: bringing back sensors to a calibration facility on a regular basis, or recalibrating on the spot by a travelling reference instrument. New data driven techniques, such as Bayesian networks (e.g. Xiang et al., 2016), might offer a solution for this problem.

On the hardware side we recommend to include active ventilation to guarantee a constant air flow over the gas sensor and suppresses unwanted internal temperature changes due to heating of electronic components. To improve the $NO_2$

measurements further we recommend to include an additional low-cost ozone sensor, e.g. Ox-B431 by Alphasense. It is likely that the linear regression approach is able to resolve a significant part of the cross-sensitivity to ozone and $NO_2$. The RH sensor signal should be used more cleverly to detect and filter sudden changes in relative humidity. Adding a local data logger is also recommended, to be able to recover data for periods when the WiFi connection to the central database is lost.

**Data availability**

A complete overview of fit results for all models can be found in the supplement. The hourly Urban AirQ sensor data, calibrated in hindsight by interpolating the calibration in time between two calibration periods, can be downloaded at https://github.com/waagsociety/making-sensor.

*Competing interests*. The authors declare that they have no conflict of interest.

*Acknowledgements.* The Urban AirQ project was partly funded by a 2016 Stimulus Grant from AMS (Advanced Metropolitan Solutions). The project is also part of Making Sense, funded by European Union's Horizon 2020 research and

innovation programme. Qijun Jiang is supported by the China Scholarship Council for his Ph.D. research. The authors would like to thank Emma Pareschi from Waag Society who was responsible for the hardware development.

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

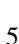

**Figure 1 Locations of the sensor devices during the citizen measurement campaign. The green marker indicates the calibration location at GGD Vondelpark. In the circle the location of SD04 and the GGD station at Oude Schans (in red). The location of Valkenburgerstraat is highlighted in yellow.**

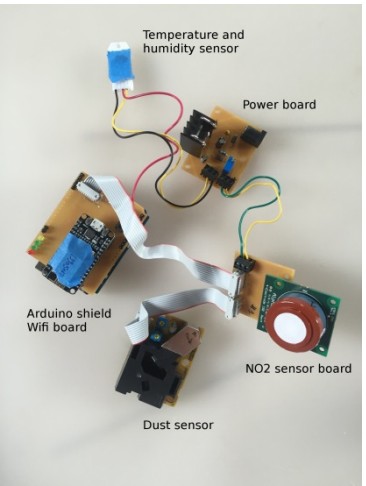
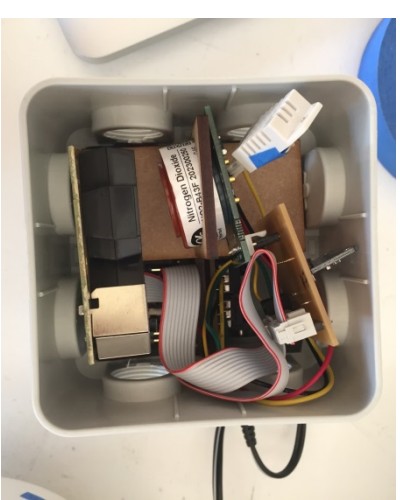
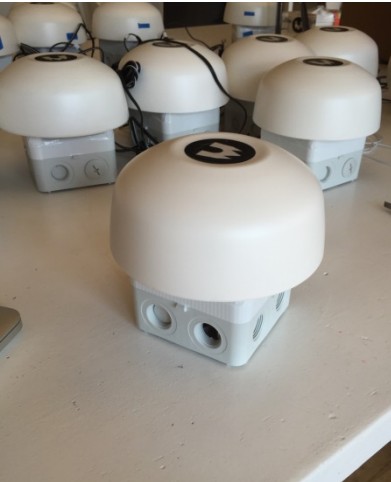

**Figure 2 Hardware modules of a sensor device (left), and the integration in the casing: open (middle) and closed (right).**

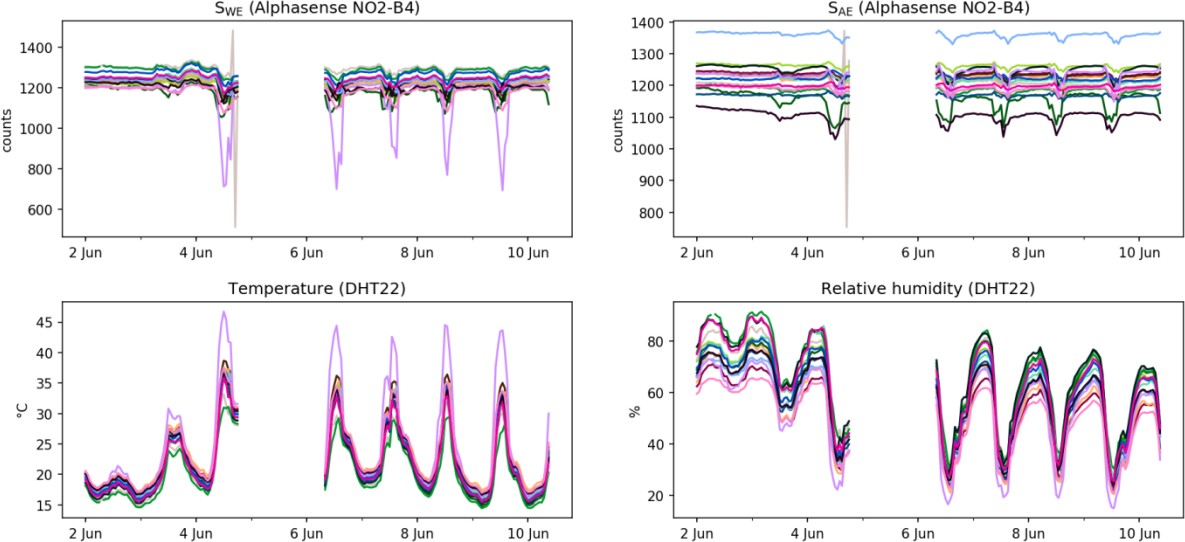

**Figure 3 Raw sensor data, unfiltered but hourly averaged, from the 16 sensors during the first calibration period, 2-10 June 2016. The data gap around 5 June is due to a connectivity problem to the central database.**

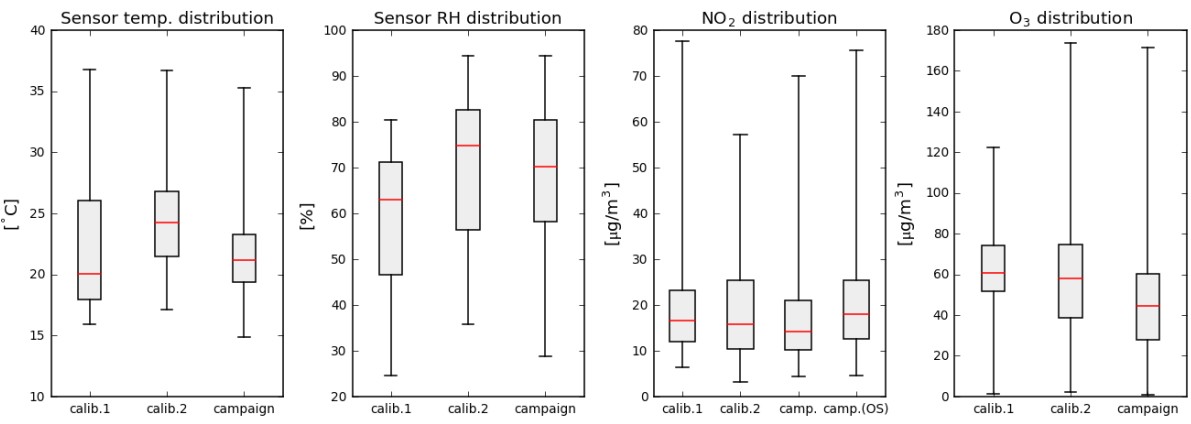

**Figure 4 Box whisker diagrams of hourly ambient parameters during the two calibration periods and the measurement campaign. The box edges indicate the $25^{th} – 75^{th}$ percentile; the whiskers the minimum and maximum values. The median is indicated in red. Temperature and RH are based on the average values of all sensors devices, $NO_2$ and ozone are taken from the reference station at Vondelpark. For comparison, $NO_2$ from the reference station at Oude Schans (OS) is also shown.**

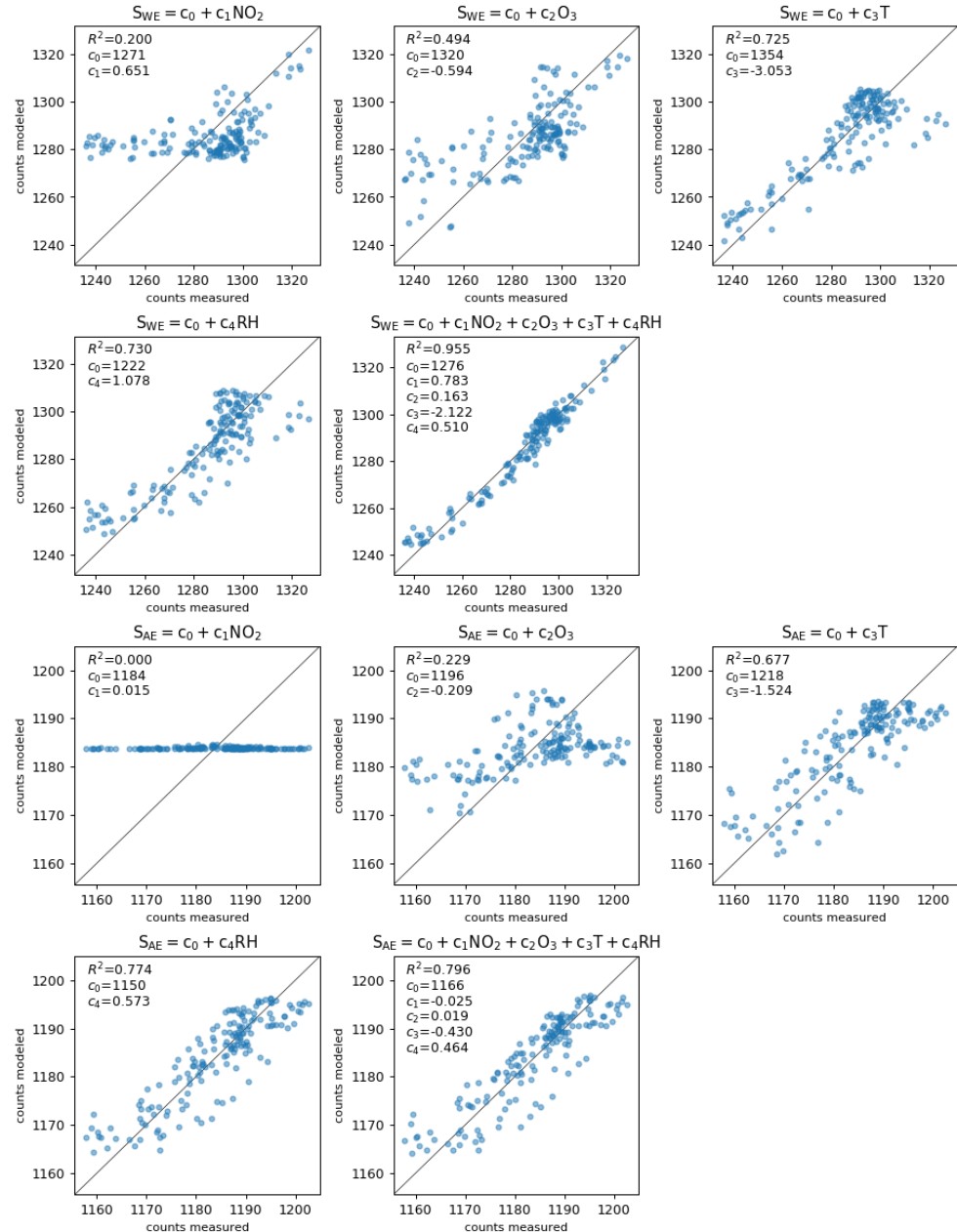

**Figure 5 The reading of a typical performing NO2-B43F sensor (SD10) explained as a linear regression of respectively $NO_2$, $O_3$, T, RH, and all variables. The top two rows show the results for the Working Electrode; the bottom two rows for the Auxiliary Electrode. The axes represent the A/D converter counts, which are proportional to the currents generated by the sensor at the corresponding electrode.**

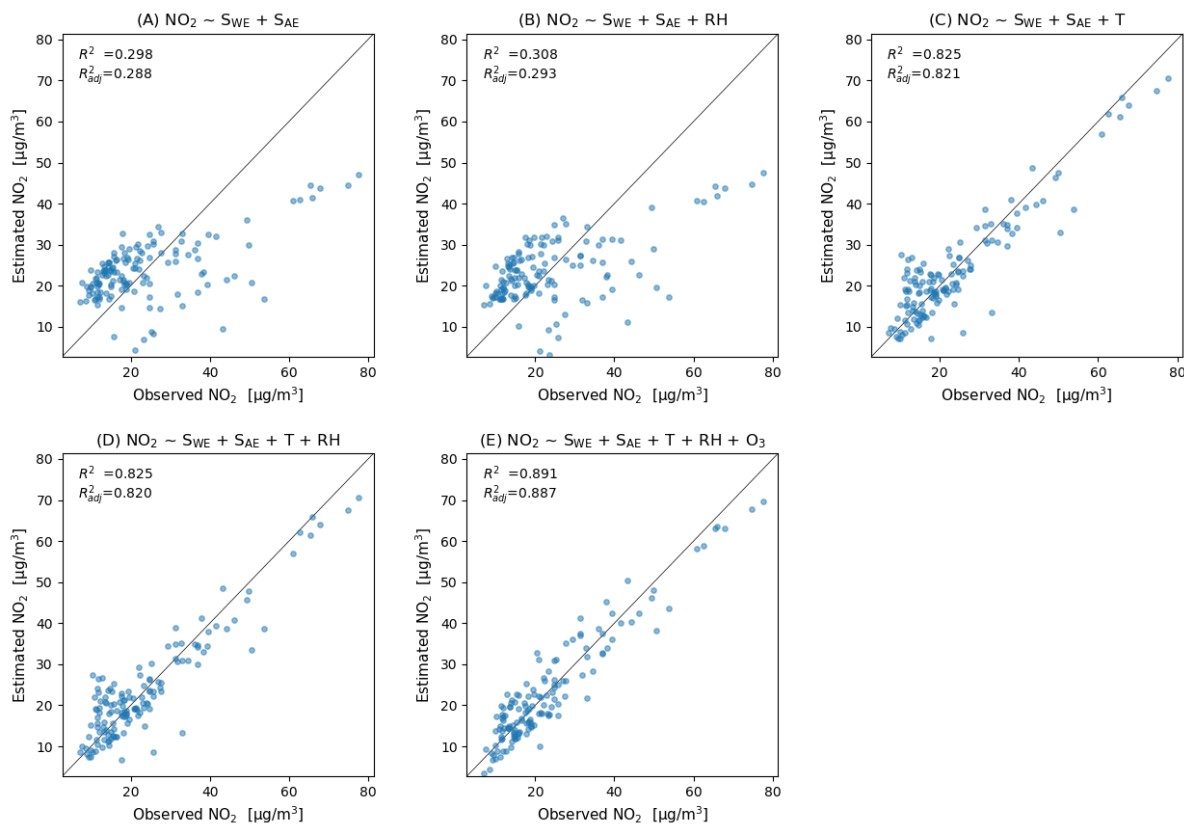

**Figure 6(a) Calibration model results for an average performing sensor (SD15). Bottom row shows the recommended calibration by Model D (left), and the results when ozone would be included (right).**

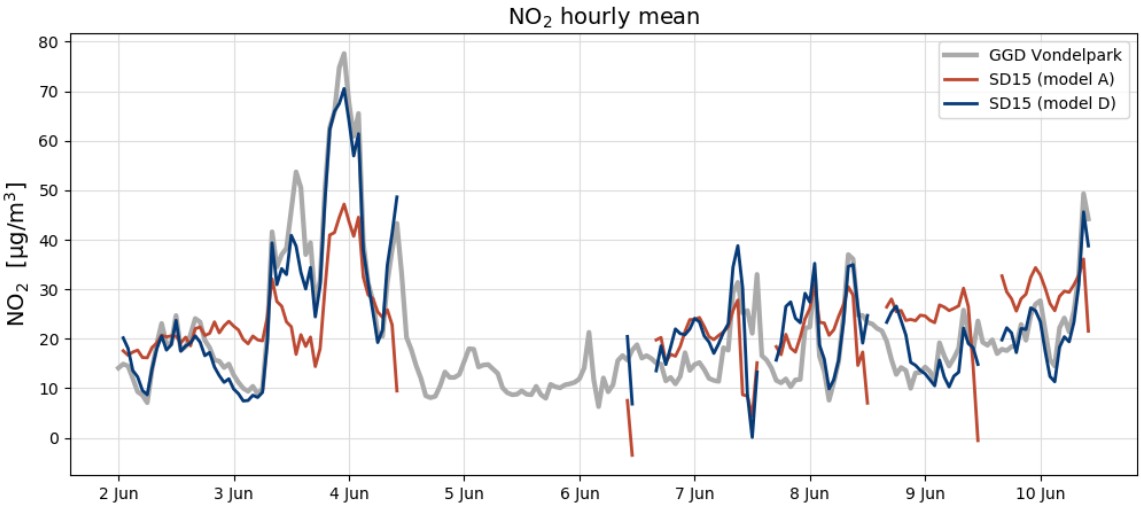

5 **Figure 6(b) Time series compared to ground truth with calibration parameters of Model A and D.**

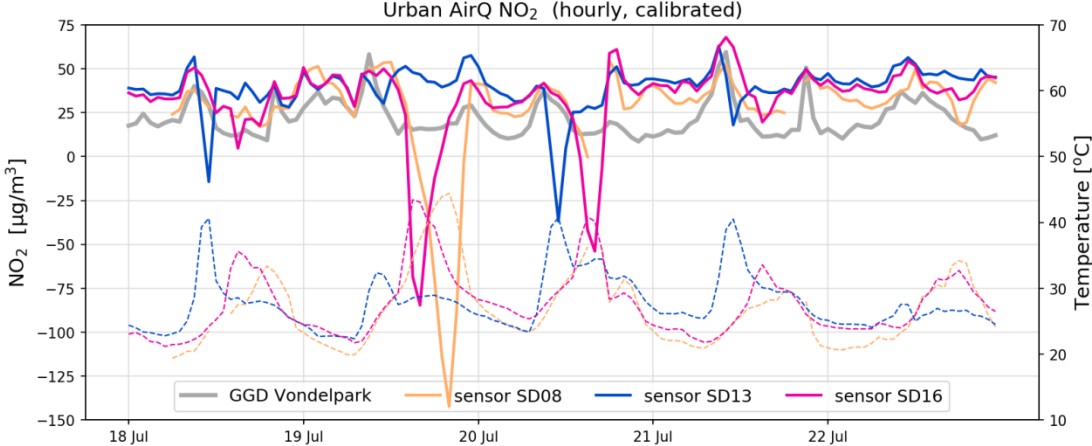

**Figure 7(a) Examples of negative spikes in the calibrated NO$_2$ measurements (solid line) due to internal sensor temperatures (dotted line) exceeding 30 °C.**

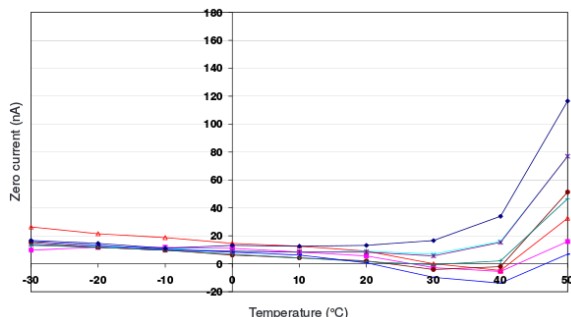

5    **Figure 7(b) Variation of zero output of the working electrode caused by changes in temperature for a typical batch of electrochemical sensors. Image taken from Alphasense Data Sheet for NO2-B43F (ADS, 2016).**

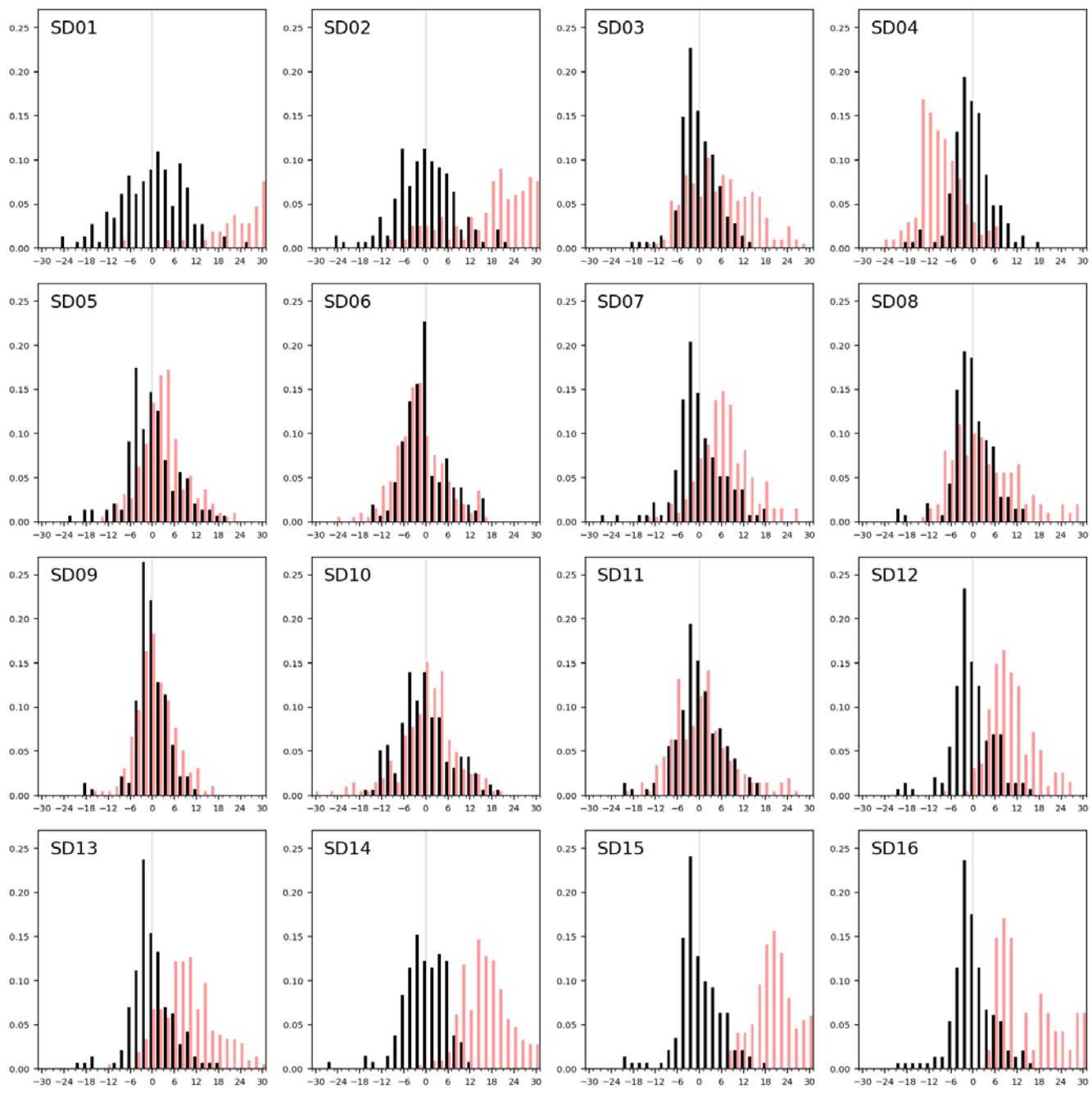

**Figure 8 Sensor drift during two months of operation, shown as the distribution of residuals (in 2 μg m⁻³ bins) with the reference measurements during the first calibration period (black bars) and during the second period (red bars).**

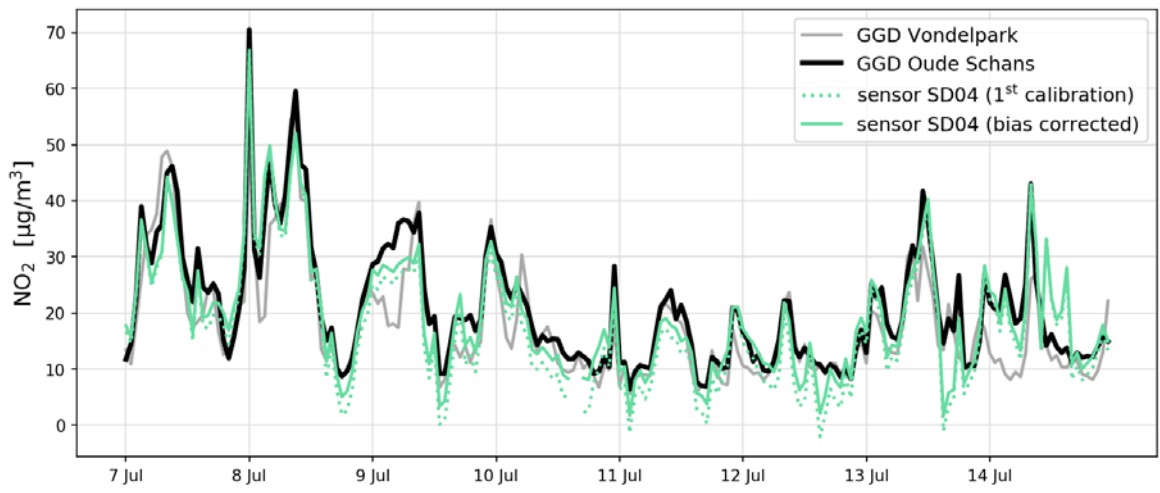

**Figure 9(a) Comparison of sensor SD04 NO$_2$ time series with the nearby Oude Schans station (8-day snap shot), and the effect of bias correction. For comparison, measurements of Vondelpark station are also shown.**

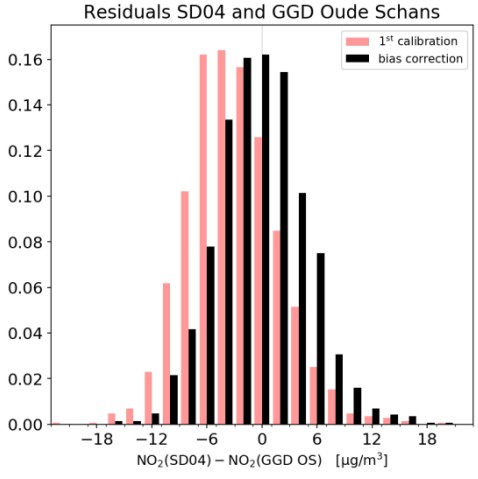

**Figure 9(b) Distribution of residuals of NO$_2$ measurements between sensor SD04 and Oude Schans station during the campaign period, with and without bias correction.**

**Table 1 Fit results for regression model A. Older NO2-B42F sensors highlighted in grey.**

| Sensor ID | $c_0$ | $c_1$ ($S_{WE}$) | $c_2$ ($S_{AE}$) | $R^2$ |
|-----------|-------|------------------|------------------|-------|
| SD01 | 455.4 | 0.6977 | -1.0835 | 0.47 |
| SD02 | 355.9 | 0.8862 | -1.2633 | 0.62 |
| SD03 | -228.6 | 1.0877 | -0.8029 | 0.72 |
| SD04 | -968.2 | 0.9138 | -0.1237 | 0.69 |
| SD05 | -155.1 | 0.8368 | -0.6841 | 0.48 |
| SD06 | -141.9 | 0.6136 | -0.5241 | 0.44 |
| SD07 | -576.4 | 0.9615 | -0.4811 | 0.57 |
| SD08 | 231.4 | 1.0802 | -1.2514 | 0.68 |
| SD09 | 100.5 | 0.8669 | -0.8952 | 0.56 |
| SD10 | 342.0 | 0.8221 | -1.1629 | 0.50 |
| SD11 | 338.4 | 0.9823 | -1.2246 | 0.61 |
| SD12 | -375.2 | 0.7775 | -0.4837 | 0.54 |
| SD13 | -1703.4 | 0.8218 | 0.5544 | 0.60 |
| SD14 | 162.6 | 0.8156 | -0.9075 | 0.46 |
| SD15 | 1211.2 | 0.9008 | -1.8984 | 0.30 |
| SD16 | -594.3 | 0.8007 | -0.3192 | 0.49 |

**Table 2 Regression models for $NO_2$**

| | | |
|---|---|---|
| Model A | $NO_2 = c_0 + c_1 \cdot S_{WE} + c_2 \cdot S_{AE}$ | Linear combination of Working Electrode and Auxiliary Electrode |
| Model B | $NO_2 = c_0 + c_1 \cdot S_{WE} + c_2 \cdot S_{AE} + c_4 \cdot RH$ | Relative humidity correction |
| Model C | $NO_2 = c_0 + c_1 \cdot S_{WE} + c_2 \cdot S_{AE} + c_3 \cdot T$ | Temperature correction |
| Model D | $NO_2 = c_0 + c_1 \cdot S_{WE} + c_2 \cdot S_{AE} + c_3 \cdot T + c_4 \cdot RH$ | Temperature and RH correction |
| Model E | $NO_2 = c_0 + c_1 \cdot S_{WE} + c_2 \cdot S_{AE} + c_3 \cdot T + c_4 \cdot RH + c_5 \cdot O_3$ | Correction for temperature, RH, and ozone cross-sensitivity |

**Table 3 Fit results for regression model D. Older NO2-B42F sensors highlighted in grey.**

| Sensor ID | $c_0$ | $c_1$ ($S_{WE}$) | $c_2$ ($S_{AE}$) | $c_3$ (T) | $c_4$ (RH) | $R^2$ |
|-----------|-------|--------|--------|--------|--------|------|
| SD01 | 790.9 | 0.8707 | -1.5645 | -0.5051 | 0.4513 | 0.62 |
| SD02 | 589.2 | 0.8618 | -1.4742 | 0.2142 | 0.4204 | 0.67 |
| SD03 | -1272.1 | 1.2045 | -0.1492 | 1.2690 | -0.2944 | 0.87 |
| SD04 | -1613.3 | 1.1499 | 0.1818 | 0.3200 | -0.4442 | 0.85 |
| SD05 | -1623.1 | 1.1235 | 0.2088 | 1.7161 | -0.4430 | 0.75 |
| SD06 | -824.8 | 1.1850 | -0.5839 | 1.6737 | -0.3069 | 0.81 |
| SD07 | -1217.6 | 1.1305 | -0.1642 | 1.9435 | 0.0000 | 0.79 |
| SD08 | -1129.7 | 1.1835 | -0.2705 | 2.2559 | -0.2704 | 0.86 |
| SD09 | -586.3 | 1.1794 | -0.6738 | 2.0415 | -0.2192 | 0.90 |
| SD10 | -1152.7 | 1.1668 | -0.3120 | 2.9112 | -0.2147 | 0.72 |
| SD11 | -1109.8 | 1.1055 | -0.2339 | 3.3191 | -0.1693 | 0.81 |
| SD12 | -1074.9 | 1.0961 | -0.2346 | 1.4954 | -0.2799 | 0.84 |
| SD13 | -1074.6 | 1.1294 | -0.3058 | 1.8671 | -0.1561 | 0.83 |
| SD14 | 8.1 | 1.1860 | -1.1889 | 2.5401 | 0.0268 | 0.84 |
| SD15 | -104.5 | 1.8111 | -1.7939 | 4.8373 | 0.0596 | 0.83 |
| SD16 | -1215.5 | 1.2551 | -0.3038 | 2.1742 | -0.1333 | 0.84 |

**Table 4 Descriptive and short-term predictive error of model D in μg m$^{-3}$**

| Sensor ID | 2-7 June (descriptive) | | 8-10 June (predictive) | |
| | Uptime | RMSE | Uptime | RMSE |
|---|---|---|---|---|
| SD01 | 92h | 9.25 | 54h | 9.31 |
| SD02 | 89h | 7.95 | 53h | 13.74 |
| SD03 | 88h | 5.58 | 53h | 4.37 |
| SD04 | 90h | 6.00 | 54h | 4.94 |
| SD05 | 90h | 7.62 | 53h | 8.75 |
| SD06 | 97h | 6.36 | 57h | 5.57 |
| SD07 | 85h | 7.09 | 52h | 6.26 |
| SD08 | 88h | 5.95 | 52h | 6.59 |
| SD09 | 88h | 4.94 | 52h | 3.69 |
| SD10 | 99h | 7.44 | 59h | 8.09 |
| SD11 | 91h | 6.78 | 53h | 5.42 |
| SD12 | 93h | 6.08 | 52h | 5.07 |
| SD13 | 89h | 6.25 | 54h | 5.31 |
| SD14 | 83h | 3.96 | 48h | 14.61 |
| SD15 | 89h | 6.75 | 52h | 4.52 |
| SD16 | 93h | 6.06 | 55h | 5.61 |

**Table 5 Bias and random error in μg m$^{-3}$ when calibrated in the first period with model D**

| Sensor ID | 1$^{st}$ calibration period | | | 2$^{nd}$ calibration period | | |
| | Uptime | Bias | SDR | Uptime | Bias | SDR |
|---|---|---|---|---|---|---|
| SD01 | 146h | -0.1 | 8.8 | 106h | 40.1 | 18.2 |
| SD02 | 142h | 0.0 | 8.2 | 199h | 21.4 | 12.8 |
| SD03 | 141h | 0.0 | 5.1 | 205h | 5.6 | 9.3 |
| SD04 | 144h | 0.0 | 5.5 | 202h | -9.2 | 5.8 |
| SD05 | 143h | 0.0 | 7.0 | 192h | 3.0 | 6.3 |
| SD06 | 154h | 0.0 | 6.0 | 197h | -2.1 | 6.8 |
| SD07 | 137h | 0.0 | 6.6 | 196h | 6.6 | 6.8 |
| SD08 | 140h | 0.0 | 5.4 | 199h | 3.1 | 9.1 |
| SD09 | 140h | 0.0 | 4.5 | 196h | 0.7 | 5.3 |

| | | | | | | |
|------|------|-----|-----|------|------|-----|
| SD10 | 158h | 0.0 | 7.2 | 206h | 0.2 | 7.9 |
| SD11 | 144h | 0.0 | 6.3 | 205h | 0.5 | 8.5 |
| SD12 | 145h | 0.0 | 5.7 | 194h | 10.1 | 6.0 |
| SD13 | 143h | 0.0 | 5.8 | 206h | 9.8 | 7.7 |
| SD14 | 131h | 0.0 | 5.9 | 211h | 16.6 | 6.9 |
| SD15 | 141h | 0.0 | 6.0 | 198h | 21.3 | 6.8 |
| SD16 | 148h | 0.0 | 5.7 | 47h | 15.6 | 8.7 |

**Table 6 Comparison of sensor SD04 with Oude Schans station during the campaign period, according to different calibrations**

| | 1$^{st}$ calibration | 2$^{nd}$ calibration | Weighted calibration |
|---|---|---|---|
| Mean $NO_2$, GGD Oude Schans | 19.96 μg m$^{-3}$ | 19.96 μg m$^{-3}$ | 19.96 μg m$^{-3}$ |
| Mean $NO_2$, sensor SD04 | 17.02 μg m$^{-3}$ | 22.21 μg m$^{-3}$ | 19.87 μg m$^{-3}$ |
| Bias | -2.94 μg m$^{-3}$ | 2.25 μg m$^{-3}$ | -0.09 μg m$^{-3}$ |
| RMSE | 6.10 μg m$^{-3}$ | 5.25 μg m$^{-3}$ | 5.20 μg m$^{-3}$ |
| Correlation | 0.89 | 0.89 | 0.88 |

