# Peer review of "Field calibration of electrochemical NO2 sensors in a citizen science context"

_Atmospheric Measurement Techniques, 2017_

## Referee Comment (RC1) · D. Ramsay (Referee) · 21 Jun 2017

First I want to say that I appreciate the hard work that goes into this. You've selected a good sensor with a good reputation, and you're methodology for a neighborhood study is at a high-level the right approach– colocation calibration, a few weeks in the field, and then colocation calibration. I think this kind of work in the citizen sensing community is important, and I'm glad that your methodology incorporates good sensor technology and recent best practice.

That said, I'm not sure what the precise contribution of this paper is.

In the realm of calibration technique and design, this is not state-of-the-art, nor is the methodology the right one if the point is the verification of a calibration algorithm. See

this paper [http://www.atmos-meas-tech-discuss.net/amt-2017-138/amt-2017-138.pdf] for an example of the latest techniques and best practice– here HDMR takes into account more complex relationships than linear dependence and more complex variable interactions. In the linked submission, superior techniques with a longer co-location periods are applied to the Alphasense NO2 sensor. Their methodology is also strong– instead of fitting their calibrations to their entire colocation dataset, they train a calibration on part of it and validate it on a holdout set. This is the proper methodology if your contribution is about multilinear calibration for electrochemical sensors. You cite many good sources that have done work of similar complexity/characterization for calibration and colocation analysis of these sensors.

I presume the intended contribution has more to do with the installation/campaign and data collection *between* co-located calibration, but I have some reservations here as well. While I do believe your data is likely reasonable given the calibration process/sensor selection/hour averaging, you haven't provided strong evidence to substantiate this belief, other than anecdotal evidence about one sensor located near another reference device. You yourself only make weak claims that it is 'good enough to detect unexpected hot spots between stations'. You also allude to the fact that (1) your colocation measurement has a lower normal ambient NO2 level than your campaign area, and (2) you don't measure O3 in your campaign area though it more strongly affects your measurement signal than NO2. This combination of facts leaves me quite concerned– the ratio of NO2/O3 might be consistent in your calibration area, and slightly different in your campaign area, and leave you with a systematic bias that you haven't properly accounted for. I don't think assuming the relative contribution of these two components is constant when you know that NO2 levels are different in the campaign area is a safe/fair assumption. The 'sudden and unexplained' offset in the only sensor you kept colocated with your reference is also slightly concerning, and deserves more explanation/treatment than your paper provides.

There are many papers published that look at citizen science installations like this,
and present novel work in other regards– things like spatio-temporal models that are validated against slightly better reference devices ('AirCloud', Sensys 2014), interesting UI for citizen interaction ('HazeWatch', Sensys 2013), etc. They are generally explicit about their contribution as a user interaction or have a slightly more compelling story around validation of their campaign data. They are also typically in human-interaction focused conferences.

I'm not convinced that having a citizen campaign by itself warrants a publication, though it forms a strong foundation to experiment/build work on top of. I do commend you on the open-sourcing of your data, and I think perhaps there is a case to be made that this aspect of it is worth publishing, but I'm still a little wary that validation of your data and key assumptions should be a little tighter (that NO2/O3 in your calibration/measurement region are similar, that your calibration technique is the proper one in the location of your measurement, etc). The lack of quantification of error in the locations you are measuring and the weak/qualitative claims about usefulness of the data are also a little disconcerting in this regard.

Finally, there are several grammatical issues floating around the paper. To me it is noticeably written by someone with English as their second language (it's quite a good job for a second language, but it's still noticeably makes it difficult to read in sections). I'd recommend going over it with a native speaker. A few obvious phrases from the paper that are not properly formed:

'Since a few years new' 'reaching more frequently higher NO2 values' 'Instead of taking from ambient air measurements' 'the corresponding correlation with true NO2 signal' 'The Korte Koningsstraat characterizes as a side street' 'sensor are more closely located' - verb tense 'most sensors have been drifting in the intermediate two-month period' - verb tense 'Alphasense NO2-B series, are known' - weird comma

This is by no means a comprehensive list, just a few I just skimmed over. More in depth grammatical review is definitely required.

I hope this work serves as a jumping off point for you to dig a little deeper into calibration technique, network validation, and/or user-facing design of air quality systems. I'm sure you'll be able to find some very interesting work to do with the data you've collected. Best of luck!

———————————————

---

## Referee Comment (RC2) · Anonymous Referee #2 · 24 Jun 2017

The paper attempts to describe a way to use low-cost electrochemical NO2 sensors (Alphasense) to extract meaningful information about ambient levels of NO2 in an urban area. Data were collected from several identical NO2 sensors (and 2 from a previous generation of Alphasense NO2 sensors) using a co-locate/measure/co-locate. Decoupling the interference between NO2 and O3 with Alphasense sensors is a difficult task, as highlighted throughout the literature. However, after reading through the manuscript several times, it does not appear the author's goal was accomplished based on their thesis: to describe a "practical method for in-field calibration and regression modeling" of electrochemical NO2 sensors. Several major concerns including the use of a reference instrument (ozone) as an independent variable within the model and lack of rigorous validation data must be addressed.

The best-performing model includes data from a reference ozone monitor which does not constitute a "practical method" for using low-cost NO2 sensors, and the regression modeling nearly completely describes how well these sensors performed in the past, without properly withholding validation data to describe how they will hold up in the future (predictive versus descriptive modeling). The modeling approach (multivariate linear regression using WE and AE) is not novel in the literature concerning Alphasense electrochemical sensors, especially when considering species other than NO2 (see Lewis 2015[1]) as an example that uses both linear regression and other statistical models).

In addition to a few major corrections, many minor corrections should be addressed as well (outlined below). Therefore, publication of this manuscript in AMT should only be considered after the comments below have been addressed.

Major Comments

P. 6, line 24: Including a reference ozone measurement as an independent variable in the linear model is inappropriate for low-cost sensing. If the goal is to describe a method by which you can use low-cost NO2 sensors to obtain a decent NO2 concentration, then including data from a $5000+ instrument in the analysis simply cannot be included. I understand that there is a strong cross-sensitivity to ozone, but claiming even a poor ozone measurement would improve results without any evidence to support the claim is invalid. This should be removed completely from the analysis.

To show the model is predictive (rather than descriptive), previously withheld validation data should be used to evaluate the model. Currently, this work only shows that these sensors can reasonably describe what has been measured in the past, but provides no insight into well they will hold up in the future.

All fit parameters in the tables (and throughout the paper) should have error estimates/confidence intervals.

A focus on the absolute RMSE, rather than just the bias-corrected RMSE should be highlighted in the abstract

Minor Comments

There are many English language errors (mostly grammar) that need to be worked out

P. 2, line 6: These sensors are commercial, not experimental, despite their quality. Stating otherwise supports the idea that they are not currently on the market, which they are.

P. 4, lines 3-4: Rather than just throwing away data based on arbitrary filters, a digital filter could be used. Throwing away data that is not within 10% of the mean is probably not the best methodology; one gives up the ability to measure higher concentrations if a local source were to emerge!

If the analysis is going to be based on the "more linear" regime of these sensors (dropping all data > 30C), it should be more pronounced in the abstract and introduction (page 4, lines 5-6). This is a huge limitation and one of the most important research topics for electrochemical sensors (as used for ambient measurements).

P. 6 line 10: If the DHT22 sensor does not need to be individually calibrated, the authors should explain why they observed such large variance between DHT22 sensors and how this affects their model results

P. 6, line 15: Comments suggest the sensor loses sensitivity at higher temperatures. This seems counterintuitive given that diffusion across the membrane should be faster at higher T. What is the explanation for this effect?

P. 11, lines 5-6: Diverging results for two different models of Alphasense NO2 sensors are discussed; Alphasense explains why the newer version of the sensor obtains better selectivity towards NO2 and has a reference (Hossain 2016 [2]) that should be examined/discussed.

Equations 6, 7: A time-based interpolation for back-calculation of NO2 is used without sufficient evidence the decay in sensitivity/accuracy is linear in time.

P. 9, line 22: Is there a reason the authors decide to use r2 rather than adjusted-r2 for comparing to adjusted-r2?

The median value throughout the campaign is 15 ugm-3 and the stated 95% CI is 14 ugm-3 (2*RMSE); what is signal and what is noise?

What makes a measurement "good enough" (page 10, line 15)?

Claiming the calibration period should be "as long as possible" isn't very helpful. Eventually, the sensitivity of the sensor would begin to decay and one would lose valuable time to move the device and measure other places! Is there a quantitative way to phrase "as long as possible"?

The description of the in-field co-location (when an NO2 sensor is compared to a close-by reference sensor) is quite confusing. It took several read-throughs to really understand when and where everything was taking place. This could be greatly simplified by adapting the map figure with notes.

P. 9, lines 6-14: The authors claim an in-field co-located NO2 sensor stays calibrated at another site, but the error bars on those measurements are the same as the absolute value of those measurements. How can one be sure they are not just looking at noise?

Figure Comments

Each figure should be able to stand alone and tell a story; many of the figures do not contribute substantially to the paper and could be omitted. Specific comments include:

Figure 1 needs labels for the co-location stations (text) to make it easier to understand what was taking place

Figure 3 demonstrates a large absolute error on some of the RH and T measurements (15 C swing on Temp and 20% on RH). Why? Should counts be converted to volts to

ease comparisons with existing literature? What is going on with the clear outlier?

Figure 5 should have more descriptive axis labels – using just the title to describe the plot makes it hard for the reader to understand what is going on. Many of these plots are not needed ([row 2, col 2], [row 3, col 1], [row 3, col 2], [row 4, col 3]). The authors claim ozone is correlated with AE response, but clearly, that is just a temperature effect. Otherwise, the authors need to describe how ozone can diffuse across the analyte and undergo a redox reaction at the AE surface.

Figure 6 is not needed. It does not add anything to the paper and is well known through basic photochemistry.

Figure 7a should not include the model with ozone in the regression (row 2, col 2)

Figure 8a does not do a good job at conveying the point (that transient temperature spikes affect the signal) since temperature is not shown anywhere.

Figure 8b is not needed. These details are in the technical spec sheet and previous literature – just cite those.

Figure 9 is not needed – simply describing the start-up/warm-up period in the methods section along with other filtering methodology was sufficient.

Figure 10a and 10b do not seem to convey what you are trying to convey – plotting a distribution of the residuals during the two co-location periods would be much more helpful.

Figure 11 was already described in a Table – no need for a plot as well. They are very confusing and don't add anything in terms of advancing the story. It just makes it seem like the linear model is not very robust or repeatable. It also appears to suggest the is a drift in the y-intercept of nearly 1000 ugm3 in some instances!

Figure 12b could also be plotted as a distribution of residuals – one would then be able to see clear overlap (or not) if there is/isn't bias.

[Figure]

[1]: http://xlink.rsc.org/?DOI=C5FD00201J [2]: http://pubs.acs.org/doi/full/10.1021/acssensors.6b00603

**AMTD**

---

## Referee Comment (RC3) · Anonymous Referee #3 · 30 Jun 2017

General Comments:

1. The work presents the process involved in trying to calibrate a low-cost NO2 sensor for citizen science work. The sensor was collocated near a regulatory monitor for a period of 6 days, deployed in a community for 2 months, and then collocated again for a period of about 9 days. The work explored a number of calibration equations and determined that the best calibration equation would consider the temperature and relative humidity influences and the co-sensitivity to ozone. However, the sensors were not built to also measure ozone and thus, a calibration scheme omitting this factor was selected.

2. Unfortunately, the calibration procedure discussed is not novel or state of the art. Based on the title, I expected that it would be one or other or dynamic and easy to apply

on the fly in the field. This definitely doesn't fit the bill. I think the manuscript would be better received if it were refocused to include a look at the data from the 2-month citizen science deployment.

3. I agree with the comments already posted by other reviews/researchers and have tried simply to add additional information in this review.

Specific Comments:

1. P4, Line 7 – Why was this criteria chosen? 33% of an hour seems rather low and at best arbitrary.

2. P4, Line 14 – Why was the collocation effort conducted at Vondelpark (urban background) and not Oude Schans (urban)? This might have minimized the differences between the calibration and study periods.

3. P4, Line 25 – Include the average deployment period/time to the citizen campaign discussion.

4. P4, Line 33 – Throughout this manuscript, be more specific about your descriptors like higher and better. Discuss the metric used to make those determinations. For examples, regarding temperature on this line, the absolute highest temperature nor the mean temperature appears to be higher during both calibration periods so what metric are you looking at?

5. P6, Line 18 – These two paragraphs should be re-visited to try to simplify. The model letters are not in order of best fit and that might help.

6. P6, Line 26 – Why is ozone considered as a metric if it wasn't routinely measured during the campaign? It reinforces your argument that it should be measured but it's really no good to you in your current work. To really lend weigh to your argument that ozone should also be measured if using this NO2 sensor, you should explore whether the ozone concentrations from the nearest monitor would be a helpful addition and if a sensor based ozone measurement is good enough to help.

7. P6, Line 30 – Discuss the technical differences between these sensor models.

8. P7, Line 2 – Use a statistical measure rather than a figure of demonstrate improved performance.

9. P7, Line 5 – What does calibrated but uncorrected mean?

10. P7, Line 13 – What factors do you think affect the stabilization time. You mention 'most' sensors stabilized within this time. How many is most? Why not provide a range? What was different about the outliers?

11. P7, Line 26 – Aging of temp and RH sensor is not widely reported as a problem. I realize the sensor was measuring in-box temperature and RH rather than ambient but is there really no available data (nearby temp and RH station) by which to but some bounds on this potential affect. Are you considering testing that hypothesis?

12. P10, Line 17 – I think it might also be worth noting what this method would not be able to detect like transient spikes from nearby sources (because you are eliminating any spike outside of 10% of the mean). Because of this exclusion criteria, why do you think you could use this model to provide realistic estimates of peak values?

13. Figure 1 – I would like to see the Vondelpark station on this map to better appreciate the distance and variation in the urban environment. It would also help to see how large of an area this study area is in comparison with the city of Amsterdam.

14. Figure 2 – Rather than the photo of the sensor boxes charging, I think it would be helpful to see how they sit within this housing to better understand the appropriateness of the temperature and relative humidity measurement, etc.

15. Figure 3 – it appears that one sensor, in particular, appears to be an outlier in most of this figures. Was its removal from the study ever considered? Why/why not?

16. Figure 4 – Please check the text to make reference to Vondelpark and Oude Schans (OS) more consistent and clear. I believe at one post one of the stations is just
referred to as GGD.

17. Figure 6 – Graph is not needed, equation and R2 in the text is sufficient for making this point.

18. Figure 7b – A Figure is not the best way to support the assertion that improved performance is clearly shown. It appears to me to be true only about 50% of the time from this figure.

19. Figure 8a – I would remove this Figure. If you leave it, include temperature.

20. Figure 8b – Just reference the data sheet.

21. Figure 9 – Figure, in this format not needed. If you want a figure, it would more useful to show error between measurements vs. time and for each sensor as it starts.

22. Figure 10 – Using similar scales would help illustrate the drift.

23. Figure 11 – Error bars/estimates for the coefficients before and after would be a helpful comparison in this Figure.

24. Figure 12b – Present R2.

25. Tables – Find a way to visually note the older sensors by ID number.

---

## Author Comment (AC3) · 1 Dec 2017

Please see Supplement

Please also note the supplement to this comment:
https://www.atmos-meas-tech-discuss.net/amt-2017-43/amt-2017-43-AC3-supplement.pdf

---

## Author Response (AR1)

**Cover letter major revision amt-2017-43**

First we would like to thank the three referees for their time to evaluate our manuscript and provide useful comments. It helped us to restructure the paper and add additional results, and so improve the quality of the presented work. The detailed changes which lead to this mayor revision can be found in the Track Changes version below.

Outline of the most important changes:

- New title which better captures the presented study: "Field calibration of electrochemical NO2 sensors in a citizen science context".
- More focus on citizen science context of the described experiment, including more recommendations for improved follow-up experiments.
- Inclusion of predictive regression, showing that the calibration model is able to predict values on a short time scale.
- An improved analysis of the introduced bias when using the sensor devices at sites where the NO2/O3 ratio is different than at the calibration site.
- An improved analysis of the temperature readings in relation to the internal sensor temperature.
- Reduction and update of figures.
- New (better readable) labelling of sensor IDs.
- Revision of the English grammar throughout the manuscript. We are willing to do a stricter language check by native speakers if the paper is selected for publication.

**Table of contents**

**Response to Referee #1, amt-2017-43**

*First I want to say that I appreciate the hard work that goes into this. You've selected a good sensor with a good reputation, and you're methodology for a neighborhood study is at a high-level the right approach– colocation calibration, a few weeks in the field, and then colocation calibration. I think this kind of work in the citizen sensing community is important, and I'm glad that your methodology incorporates good sensor technology and recent best practice. That said, I'm not sure what the precise contribution of this paper is.*

New low-cost sensor technology for air quality application is available for several years now, and is used in many experiments often done by motivated but not necessarily scientifically trained people. This can result in gathering of data which, due to their poor quality, is unusable for quantifying air pollution. Our study shows that, if proper attention is payed to calibration, such experiments with low-cost sensors can result in useful measurements.

In its first submission, however, the paper focussed more on the technicalities of the calibration we applied, which might have confused the reader (or reviewer) that we are dealing with a strict scientific experiment in which all variables can be controlled. On the contrary, as our study deals with data which is generated in a citizen science campaign, one has to be creative to make sense of the gathered data.

Therefore, we have shifted the focus to how to deal with the analysis of air quality data which is collected with imperfect sensors under imperfect conditions (e.g. in a citizen science campaign). We will explain our calibration approach, but put more attention to our lessons learnt and recommendations on hardware, experimental set-up, and data analysis approach, as we believe that many future campaigns will benefit greatly from this information. This is now reflected in the new title "Field calibration of electrochemical $NO_2$ sensors in a citizen science context". We left the "Practical" out, as the sensor degradation issue prevent a really practical calibration scheme which can be used for similar initiatives.

*In the realm of calibration technique and design, this is not state-of-the-art, nor is the methodology the right one if the point is the verification of a calibration algorithm. See this paper [http://www.atmos-meas-tech-discuss.net/amt-2017-138/amt-2017-138.pdf] for an example of the latest techniques and best practice– here HDMR takes into account more complex relationships than linear dependence and more complex variable interactions. In the linked submission, superior techniques with a longer co-location periods are applied to the Alphasense NO2 sensor.*

The mentioned paper, Cross et al. (2017), was submitted to AMT on April 28, while our paper was submitted more than two months earlier. HDMR might be a more sophisticated method than the widely understood linear regression method used in our study. For $NO_2$, however, the authors find a RMSE of 8.6 µg/m$^3$ (4.56 ppb) for their test data, which is comparable to our estimated 7 µg/m$^3$ when applying our weighted calibration method based on multilinear regression. Unfortunately, Cross et al. do not give insight in their optimal HDMR model for

NO$_2$ nor the sensitivity indices for input pairs (maybe because of the propriety nature of the ARISense device?); it remains unrevealed which signal relation best describes the NO2 concentration in their study.

They use training data which is distributed over a 4.5 month interval to derive a calibration model.

Given the fact that all devices must be calibrated individually, this is an impractical long period before they can
5  be deployed at locations where no reference data is available. Furthermore, by using training data which is distributed over the entire period, sensor degradation within that period cannot be detected. Our study shows that, using training data from a consecutive period, degradation during a successive multiple-month period is significant.

*Their methodology is also strong– instead of fitting their calibrations to their entire colocation dataset, they*
10  *train a calibration on part of it and validate it on a holdout set. This is the proper methodology if your contribution is about multilinear calibration for electrochemical sensors.*

In the revised version, we included a predictive analysis in which the calibration is based on the first half of the calibration period, and the second half of this period is used for validation. The results show that the regression model describes well the measurements on short term, but loses predictability on the long term (e.g. two
15  months) due to sensor degradation.

*I presume the intended contribution has more to do with the installation/campaign and data collection **between** co-located calibration, but I have some reservations here as well. While I do believe your data is likely reasonable given the calibration process/sensor selection/hour averaging, you haven't provided strong evidence to substantiate this belief, other than anecdotal evidence about one sensor located near another reference device.*
20  *You also allude to the fact that (1) your colocation measurement has a lower normal ambient NO2 level than your campaign area, and (2) you don't measure O3 in your campaign area though it more strongly affects your measurement signal than NO2. This combination of facts leaves me quite concerned– the ratio of NO2/O3 might be consistent in your calibration area, and slightly different in your campaign area, and leave you with a systematic bias that you haven't properly accounted for. I don't think assuming the relative contribution of these*
25  *two components is constant when you know that NO2 levels are different in the campaign area is a safe/fair assumption.*

We believe that the good agreement of sensor 54200 with the readings of an independent reference station OS (located at 3 km distance from the calibration site at Vondelpark) is more than anecdotal evidence. As can be shown in Table 5, RMSE of this sensor is 5.2 µg/m$^3$ during the two-month campaign period. From Figure 4 can
30  be seen that ozone levels were generally lower than during the calibration period, but still the bias is acceptably small (-0.09 µg/m$^3$) meaning that the collinearity between temperature and ozone holds for both locations.

It must be said that both OS and Vondelpark station qualify as a city background station, which implies that they have similar NO$_2$/O$_3$ ratios. The Referee is right in his concern about the influence of different NO$_2$/O$_3$ ratios

found at locations closer to emission sources. To get a better understanding of the possible impact, we compared hourly ozone measurements from the GGD authorities at Van Diemenstraat (VDS, classified as street station) against Nieuwdammerdijk (NDD, classified as urban background station) during June-August 2016. The location of these stations can be found at [www.luchtmeetnet.nl](http://www.luchtmeetnet.nl). The relation can best be described by $[O_3]_{VDS}$ =

5  0.87 $[O_3]_{NDD}$ + 0.85, which means that ozone levels at the street station are typically 13% lower that at the background station, due to titration of $O_3$ with NO. As the electrochemical $NO_2$ sensor is cross-sensitive for ozone, larger values must be subtracted from the sensor signal when the ozone concentration increases. This explains the negative ozone coefficient $c_5$ we find with calibration model E. According to the regression results in the Supplement a typical value for $c_5$ is -0.3. Calibration with model D will overcorrect (i.e. subtract too much)

10  for locations which have lower ozone concentrations than at the calibration site, resulting in an underestimation of $NO_2$ concentrations. For $[O_3]$=60 µg/m$^3$ (75 percentile of the distribution during the measurement camping, according to Figure 4) we estimate the underestimation in $NO_2$ at street side as 0.3 × 13% × 60 = 2.3 µg/m$^3$.

We included this elaboration on location dependency of the calibration model now in the Discussion section. As

15  already indicated in the Conclusions and Outlook, we believe that the inclusion of an additional low-cost ozone sensor (e.g. Ox-B431 by Alphasense) in an updated version of the device will reduce the bias due to different $NO_2/O_3$ ratios at different locations.

*The 'sudden and unexplained' offset in the only sensor you kept colocated with your reference is also slightly concerning, and deserves more explanation/treatment than your paper provides.*

20  We further analyzed the data of the reference sensor (55303) and found the cause of this sudden jump. Initially, this device not equipped with a PM10 module. Half-way the campaign, the technical operators decided to add this module, and removed the sensor between 10 and 14 July for service. Once placed back, temperature measurements by its DHT-22 sensor show that the internal device temperature increased by 2.5 degree on average. This can be attributed to increased power dissipation: after the periodic WiFi connection (350 mA

25  peak), the PM module is the largest consumer of electricity (80 mA). This sudden jump in temperature is the main cause for the disrupted reference series. This is now included in Section 4.6.

*There are many papers published that look at citizen science installations like this, and present novel work in other regards– things like spatio-temporal models that are validated against slightly better reference devices ('AirCloud', Sensys 2014), interesting UI for citizen interaction ('HazeWatch', Sensys 2013), etc. They are*

30  *generally explicit about their contribution as a user interaction or have a slightly more compelling story around validation of their campaign data. They are also typically in human-interaction focused conferences.*

More focus has been put on the citizen science aspect of our experiment. Unlike the mentioned projects, our study focusses on low-cost NO2 sensing, which due to its specific calibration issues needs special attention to be successfully applied in a citizen science setting.

*I'm not convinced that having a citizen campaign by itself warrants a publication, though it forms a strong foundation to experiment/build work on top of.*

The revised paper is now stronger on the 'lessons learnt' side, so that the paper also can be read as a guide for more successfully setting up similar citizen science campaigns.

*I do commend you on the open-sourcing of your data, and I think perhaps there is a case to be made that this aspect of it is worth publishing, but I'm still a little wary that validation of your data and key assumptions should be a little tighter (that $NO_2/O_3$ in your calibration/measurement region are similar, that your calibration technique is the proper one in the location of your measurement, etc). The lack of quantification of error in the locations you are measuring and the weak/qualitative claims about usefulness of the data are also a little disconcerting in this regard.*

We feel that the inclusion of new analyses in the revised paper adds to the validity of our results. However, the set-up of the experiment limits the possibilities to answer some of these questions in detail at this stage, but it gives us directions how to better organize future experiments.

*Finally, there are several grammatical issues floating around the paper. (…) More in depth grammatical review is definitely required.*

Thank you for pointing this out. We revised the grammar throughout the paper. We are willing to do a stricter language check by native speakers if the paper is selected for publication, and it is still considered necessary.

**Response to Referee #2, amt-2017-43**

*Decoupling the interference between NO2 and O3 with Alphasense sensors is a difficult task, as highlighted throughout the literature. However, after reading through the manuscript several times, it does not appear the author's goal was accomplished based on their thesis: to describe a "practical method for in-field calibration and regression modeling" of electrochemical NO2 sensors. Several major concerns including the use of a reference instrument (ozone) as an independent variable within the model and lack of rigorous validation data must be addressed.*

*The best-performing model includes data from a reference ozone monitor which does not constitute a "practical method" for using low-cost NO2 sensors, and the regression modeling nearly completely describes how well these sensors performed in the past, without properly withholding validation data to describe how they will hold up in the future (predictive versus descriptive modeling). The modeling approach (multivariate linear regression using WE and AE) is not novel in the literature concerning Alphasense electrochemical sensors, especially when considering species other than NO2 (see Lewis 2015[1]) as an example that uses both linear regression and other statistical models).*

New low-cost sensor technology for air quality is available for several years now, and is used in many experiments often done by motivated but not necessarily scientifically trained people. This can result in gathering of data which, due to their poor quality, is unusable for quantifying air pollution. Our study shows that, if proper attention is payed to calibration, such experiments with low-cost sensors can result in useful measurements.

In its first submission, however, the paper focussed more on the technicalities of the calibration we applied, which might have confused the reader (or reviewer) that we are dealing with a strict scientific experiment in which all variables can be controlled. On the contrary, as our study deals with data which is generated in a citizen science campaign, one has to be creative to make sense of the gathered data.

Therefore, we have shifted the focus to how to deal with the analysis of air quality data which is collected with imperfect sensors under imperfect conditions (e.g. in a citizen science campaign). We still explain our calibration, but put more attention to our lessons learnt and recommendations on hardware, experimental setup, and data analysis approach, as we believe that many future campaigns will benefit greatly from this information. This is now reflected in the new title "Field calibration of electrochemical NO2 sensors in a citizen science context". We left the "Practical" out, as the sensor degradation issue prevent a really practical calibration scheme which can be used for similar initiatives.

*In addition to a few major corrections, many minor corrections should be addressed as well (outlined below). Therefore, publication of this manuscript in AMT should only be considered after the comments below have been addressed.*

*Major Comments*

*P. 6, line 24: Including a reference ozone measurement as an independent variable in the linear model is inappropriate for low-cost sensing. If the goal is to describe a method by which you can use low-cost NO2 sensors to obtain a decent NO2 concentration, then including data from a $5000+ instrument in the analysis simply cannot be included. I understand that there is a strong cross-sensitivity to ozone, but claiming even a poor*

5  *ozone measurement would improve results without any evidence to support the claim is invalid. This should be removed completely from the analysis.*

Cross-sensitivity to ozone is an important sensor issue, and should be corrected for to get more accurate low-cost NO2 measurements. We think it is appropriate to include it in the analysis to get a better understanding of cross-sensitivity to ozone. We show that the accuracy of the low-cost measurements increase when ozone is

10  included in the correction. This does not mean that the sensor devices should be equipped with a $5000+ instrument. We soften our claim that the performance of the device will improve significantly when low-cost ozone sensors are included (Section 6): "To improve the $NO_2$ measurements further we recommend to include an additional low-cost ozone sensor, e.g. Ox-B431 by Alphasense. It is likely that the linear regression approach is able to resolve a significant part of the cross-sensitivity to ozone and $NO_2$." .

15  *To show the model is predictive (rather than descriptive), previously withheld validation data should be used to evaluate the model. Currently, this work only shows that these sensors can reasonably describe what has been measured in the past, but provides no insight into well they will hold up in the future.*

Good point. We included a predictive analysis in Section 4.6 in which the calibration is based on the first half of the calibration period, and the second half of this period is used for validation. The results show that the

20  regression model describes well the measurements on short term, but loses predictability on the long term (e.g. two months) due to sensor degradation.

*All fit parameters in the tables (and throughout the paper) should have error estimates/confidence intervals.*

The standard deviations of the regression coefficients are now included in the tables of the Supplement.

*A focus on the absolute RMSE, rather than just the bias-corrected RMSE should be highlighted in the abstract*

25  Our claim that "the standard deviation of a typical sensor device for $NO_2$ measurements was found to be 7 µg m$^{-3}$" in the Abstract is based on  the assumption that the weighted calibration approach (described in Section 4.7) removes the sensor bias largely, which is supported by our findings in Section 4.8.

**Minor Comments**

*There are many English language errors (mostly grammar) that need to be worked out*

30  We revised the grammar throughout the paper. We are willing to do a stricter language check by native speakers if the paper is selected for publication, and it is still considered necessary.

*P. 2, line 6: These sensors are commercial, not experimental, despite their quality. Stating otherwise supports the idea that they are not currently on the market, which they are.*

Adjusted to "many low-cost air quality sensors suffer from various technical issues which limit their applicability."

5   *P. 4, lines 3-4: Rather than just throwing away data based on arbitrary filters, a digital filter could be used. Throwing away data that is not within 10% of the mean is probably not the best methodology; one gives up the ability to measure higher concentrations if a local source were to emerge!*

This filter criteria was selected after carefully studying the raw (1-minute) data. As can be seen in Figure 3, the +- 10% bandwidth is wide enough to contain all valid measurements in the linear regime. The filter criterion is
10  simple, yet effective. We did tests with advanced noise filtering using a Fourier transform, but this did not result in significant improvement of the hourly data quality. Added to the text: "This criterion was used for its simplicity and effectiveness. Note that, due to the large offset in the raw $S_{WE}$ and $S_{AE}$ signal, realistic $NO_2$ peak values are still detectable as the corresponding sensor response is still within a 10% bandwidth."

*If the analysis is going to be based on the "more linear" regime of these sensors (dropping all data > 30C), it*
15  *should be more pronounced in the abstract and introduction (page 4, lines 5-6). This is a huge limitation and one of the most important research topics for electrochemical sensors (as used for ambient measurements).*

This is now included in the abstract: "Using our approach, the standard deviation of a typical sensor device for $NO_2$ measurements was found to be 7 µg m$^{-3}$, *provided that temperatures are below 30°C*.". This limitation is further addressed in the Conclusion/Outlook.

20  *P. 6 line 10: If the DHT22 sensor does not need to be individually calibrated, the authors should explain why they observed such large variance between DHT22 sensors and how this affects their model results*

The spread in temperature and RH displayed in the raw data is partly explained by the sensor-to-sensor variability. By looking at nighttime temperatures (to eliminate the effect of local heating by exposure to direct sunlight) we discovered that all derived sensor temperatures are 2-5 degrees higher than the ambient
25  temperature. The devices are not actively ventilated (updating the hardware with active ventilation is now included in the recommendations!), which means that the energy dissipation of the device influences their internal temperature. The variable position of the temperature sensors with respect to these heat sources further explain the variance in temperature. This analysis is now included in the analysis of the raw measurement in Section 3.1.

30  *P. 6, line 15: Comments suggest the sensor loses sensitivity at higher temperatures. This seems counterintuitive given that diffusion across the membrane should be faster at higher T. What is the explanation for this effect?*

From the technical data sheet shown in Figure 8(b), one can see that sensitivity of the $NO_2$ sensor decreases linearly with temperature up to around 30 degrees. Above 40 degrees the sensor gains sensitivity with rising temperatures. This is now mentioned in Section 4.4. The application of a detailed temperature dependency model to describe this non-linear behavior was considered outside the scope of our research.

5  *P. 11, lines 5-6: Diverging results for two different models of Alphasense NO2 sensors are discussed; Alphasense explains why the newer version of the sensor obtains better selectivity towards NO2 and has a reference (Hossain 2016 [2]) that should be examined/discussed.*

Loss of sensitivity during lifetime and improved sensor design are now mentioned in Section 4.3: "The two worst performing sensor devices (SD02 and SD01) contain the older NO2-B42F sensor. The newer NO2-B43F
10  model is designed to have higher sensitivity to NO2 and less interference of ozone. The old sensor model has indeed smaller coefficients for $S_{WE}$ and larger correction terms for ozone (see the $c_1$ and $c_5$ coefficients of model E in the Supplement). This, however, can also be related to their longer operating time, as both sensors have been used in previous experiments for more than a year."

*Equations 6, 7: A time-based interpolation for back-calculation of NO2 is used without sufficient evidence the
15  decay in sensitivity/accuracy is linear in time.*

We feel that the assumption that the degradation is linear in time is the best to be made, given the limited data of our experiment and the absence of relevant scientific literature assessing electrochemical sensor degradation.

*P. 9, line 22: Is there a reason the authors decide to use r2 rather than adjusted-r2 for comparing to adjusted-r2?*

20  Thank you for pointing this out. We now include an analysis of the adjusted $R^2$ in the analysis of the $NO_2$ calibration models in Section 4.3 and in Figure 7(a). However, the adjusted $R^2$ does not change dramatically from $R^2$, as the number of observations ($n \approx 150$) is relatively high compared to the number of regression variables ($k=2...5$).

*The median value throughout the campaign is 15 ugm-3 and the stated 95% CI is 14 ugm-3 (2*RMSE); what is
25  signal and what is noise?*

From our error estimation of the sensor devices one can conclude that for low $NO_2$ values the noise dominates the signal. However, from Figure 4 can be seen that about 25% of the measurements at Oude Schans station were above 25 ug/m3 during the campaign (one is usually more interested in detecting occurrences of high pollution levels). At these levels the signal to noise is significantly better.

30  *What makes a measurement "good enough" (page 10, line 15)?*

Gradients in NO2 over the city are often too local that all features can be captured by the limited amount of official air quality stations. When looking at the difference between Vondelpark station and Oude Schans station (both classified as city background stations) between June and August 2016, 22% of the hourly measurements differ more than 7 ug/m3, and 6% of the hourly measurements differ more than 14 ug/m3. These ratios

5    increase further when considering road side stations. From this perspective, sensor devices with an accuracy around 7 ug/m3 can contribute to an improved understanding of spatial patterns.

*Claiming the calibration period should be "as long as possible" isn't very helpful. Eventually, the sensitivity of the sensor would begin to decay and one would lose valuable time to move the device and measure other places! Is there a quantitative way to phrase "as long as possible"?*

10    The Referee is right in his remark that sensor degradation would interfere with long calibration times. We changed the text to: "It is hard to quantify an optimal length of a calibration period without having a proper understanding of the sensor degradation rate beforehand. The measurement period should be at least a few days to capture the sensors behavior under a wide range of pollution levels and meteorological conditions. Very long calibration periods (in the order of months) will cause sensor degradation issues to interfere with the

15    calibration results."

*The description of the in-field co-location (when an NO2 sensor is compared to a closeby reference sensor) is quite confusing. It took several read-throughs to really under- stand when and where everything was taking place. This could be greatly simplified by adapting the map figure with notes.*

Thank you for this suggestion. We added more information on the map in Figure 1, which now should explain

20    better the set-up of our study.

*P. 9, lines 6-14: The authors claim an in-field co-located NO2 sensor stays calibrated at another site, but the error bars on those measurements are the same as the absolute value of those measurements. How can one be sure they are not just looking at noise?*

The correlation with measurements of the nearby site (Oude Schans) is 0.88 (Table 5), showing that the sensor

25    device is measuring $NO_2$ reasonably well (see also Figure 12). If we were looking at noise, correlations would be close to 0.

**Figure Comments**

*Each figure should be able to stand alone and tell a story; many of the figures do not contribute substantially to the paper and could be omitted. Specific comments include:*

30    *Figure 1 needs labels for the co-location stations (text) to make it easier to understand what was taking place*

We added more information on the map in Figure 1, which now should explain better the set-up of our study.

*Figure 3 demonstrates a large absolute error on some of the RH and T measurements (15 C swing on Temp and 20% on RH). Why? Should counts be converted to volts to ease comparisons with existing literature? What is going on with the clear outlier?*

Temperature and RH are converted from mV according to the specs of the DHT-22 sensor manufacturer. The spread in temperature and RH displayed in the raw data is partly explained by the sensor-to-sensor variability. As explained above, all derived internal sensor temperatures were found to be 2-5 degrees higher than the ambient temperature, indicating that the energy dissipation of the device influences its internal temperature. During daytime, the exposure to direct sunlight (the devices were places the rooftop of the monitoring station) contributes further to the temperature outliers seen in Figure 3. These happen in the strong non-linear regime of the NO2 sensor, which explains the corresponding strong dips in the SWE signal. This elaboration is now included in Section 3.1.

*Figure 5 should have more descriptive axis labels – using just the title to describe the plot makes it hard for the reader to understand what is going on. Many of these plots are not needed ([row 2, col 2], [row 3, col 1], [row 3, col 2], [row 4, col 3]). The authors claim ozone is correlated with AE response, but clearly, that is just a temperature effect. Otherwise, the authors need to describe how ozone can diffuse across the analyte and undergo a redox reaction at the AE surface.*

We clarified the plots by adjusting the title and including the regression coefficients. We decided not to leave out panels, as we feel that all panels illustrate a different aspect of the behavior of the sensor. However, we improved our description of this figure in the text.

*Figure 6 is not needed. It does not add anything to the paper and is well known through basic photochemistry.*

We agree. We took this plot out and explain textually.

*Figure 7a should not include the model with ozone in the regression (row 2, col 2)*

We think it is appropriate to leave it in the analysis to get a better understanding of cross-sensitivity to ozone.

*Figure 8a does not do a good job at conveying the point (that transient temperature spikes affect the signal) since temperature is not shown anywhere.*

We included a second y-axis in this plot with the internal sensor temperatures to better illustrate this non-linear temperature effect.

*Figure 8b is not needed. These details are in the technical spec sheet and previous literature – just cite those.*

We feel that this figure is illustrative for better understanding the non-linear temperature behavior, losing sensitivity with increasing temperatures, followed by a strong gain in sensitivity for higher temperatures.

*Figure 9 is not needed – simply describing the start-up/warm-up period in the methods section along with other filtering methodology was sufficient.*

We agree. We took this plot out and explain textually.

*Figure 10a and 10b do not seem to convey what you are trying to convey – plotting a distribution of the residuals during the two co-location periods would be much more helpful.*

Good suggestion. We adjusted the figure accordingly.

*Figure 11 was already described in a Table – no need for a plot as well. They are very confusing and don't add anything in terms of advancing the story. It just makes it seem like the linear model is not very robust or repeatable. It also appears to suggest the is a drift in the y-intercept of nearly 1000 ugm3 in some instances!*

We agree and took this figure out.

*Figure 12b could also be plotted as a distribution of residuals – one would then be able to see clear overlap (or not) if there is/isn't bias.*

Good suggestion. We adjusted the figure accordingly, and extended Table 5 with statistic summaries for the 1st and 2nd calibration periods.

**Response to Referee #3, amt-2017-43**

*General Comments:*

*1. The work presents the process involved in trying to calibrate a low-cost NO2 sensor for citizen science work. The sensor was collocated near a regulatory monitor for a period of 6 days, deployed in a community for 2 months, and then collocated again for a period of about 9 days. The work explored a number of calibration equations and determined that the best calibration equation would consider the temperature and relative humidity influences and the co-sensitivity to ozone. However, the sensors were not built to also measure ozone and thus, a calibration scheme omitting this factor was selected.*

We conclude that the calibration without the ozone signal gives good results e.g. from the agreement of sensor 54200 with the readings of an independent reference station located at 3 km distance from the calibration site (RMSE of 5.2 $\mu g/m^3$ and negligible bias, see Figure 12). The collinearity between temperature, RH and ozone solves part of the sensor's cross-sensitivity to ozone. We now include a discussion how this calibration generates a bias at locations where the NO2/O3 ratio deviates from the calibration site. We estimate underestimations of $NO_2$ concentrations at street sides to be smaller than 2.3 $\mu g/m^3$ 75% of the time (see response to Referee #1).

*2. Unfortunately, the calibration procedure discussed is not novel or state of the art. Based on the title, I expected that it would be one or other or dynamic and easy to apply on the fly in the field. This definitely doesn't fit the bill. I think the manuscript would be better received if it were refocused to include a look at the data from the 2-month citizen science deployment.*

In the revision, we shift the focus to how to deal with the analysis of air quality data which is collected with imperfect sensors under imperfect conditions (e.g. in a citizen science campaign). We still explain our calibration, but put more attention to our lessons learnt and recommendations on hardware, experimental setup, and data analysis approach, as we believe that many future campaigns will benefit from this information. This is now reflected in the new title "Field calibration of electrochemical NO2 sensors in a citizen science context". An in-depth analysis of the campaign data will be the subject of a following paper.

*3. I agree with the comments already posted by other reviews/researchers and have tried simply to add additional information in this review.*

*Specific Comments:*

*1. P4, Line 7 – Why was this criteria chosen? 33% of an hour seems rather low and at best arbitrary.*

This criterion was found to be a good trade-off between noise reduction by averaging and not losing too many hourly measurements. This is now included in the text.

*2. P4, Line 14 – Why was the collocation effort conducted at Vondelpark (urban back- ground) and not Oude Schans (urban)? This might have minimized the differences between the calibration and study periods.*

Both Vondelpark as Oude Schans are classified as urban background stations. Vondelpark measures a broad range of species such as NO, NO2, PM2.5 and PM10, whereas Oude Schans only measures NO and NO2.
5 Furthermore, Vondelpark station has better facilities such as accessibility, physical space, power supply, and internet connection.

*3. P4, Line 25 – Include the average deployment period/time to the citizen campaign discussion.*

Added to text: "In this 1537-hour period the devices produced 1204 valid hourly measurements on average."

*4. P4, Line 33 – Throughout this manuscript, be more specific about your descriptors like higher and better.*
*Discuss the metric used to make those determinations. For examples, regarding temperature on this line, the absolute highest temperature nor the mean temperature appears to be higher during both calibration periods so what metric are you looking at?*

Our discussion of the distributions is based on the values of the $75^{th}$ percentile. This is now included in the text.
Also added to P6, Line 16: "As the electrochemical $NO_2$ sensor loses sensitivity at higher temperatures *(see the*
15 *negative slope in Figure 7(b) for temperatures below 30°C)*"

*5. P6, Line 18 – These two paragraphs should be re-visited to try to simplify. The model letters are not in order of best fit and that might help.*

We swapped the B and C labels of the calibration models, so model A to E are now in order of increasing performance. We rewrote the mentioned paragraph to:

20 "From the fit results  we see that Model B (including RH) performs better than Model A, but Model C (including T) outperforms Model B. When both RH and T are included (Model D) the results of Model C are improved marginally. This can be understood in terms of a strong sensor dependence on temperature, a weak dependence on RH, and the collinearity between temperature and RH. Note that measuring RH is essential for guarding the data quality of electrochemical sensors, as these sensors are very sensitive to *sudden changes* in
25 RH, see e.g. AAN (2013) and Pang et al. (2016)."

*6. P6, Line 26 – Why is ozone considered as a metric if it wasn't routinely measured during the campaign? It reinforces your argument that it should be measured but it's really no good to you in your current work. To really lend weigh to your argument that ozone should also be measured if using this NO2 sensor, you should explore whether the ozone concentrations from the nearest monitor would be a helpful addition and if a sensor based*
*ozone measurement is good enough to help.*

Ozone is measured at three locations in Amsterdam: two urban background locations, and one street side location (see www.luchtmeetnet.nl). Due to the chemical lifetime of ozone (which is long compared to $NO_2$), the ozone gradients over the city are rather smooth, except in the vicinity of NOx sources (such as motorized traffic) where ozone levels are generally lower due to titration by NO. From ozone measurement during the considered three-month period we derive that this reduction in ozone is around 13% (see our response to Referee #1). The relevance of including calibration model E in our study is that it quantifies the cross-sensitivity to ozone and enables us to make an estimation of the introduced bias when the sensor devices are located at a street side. This analysis is now included in the revised text.

*7. P6, Line 30 – Discuss the technical differences between these sensor models.*

Added to Section 4.3: "The two worst performing sensor devices (SD02 and SD01) contain the older NO2-B42F sensor. The newer NO2-B43F model is designed to have higher sensitivity to NO2 and less interference of ozone. The old sensor model has indeed smaller coefficients for $S_{WE}$ and larger correction terms for ozone (see the $c_1$ and $c_5$ coefficients of model E in the Supplement). This, however, can also be related to their longer operating time, as both sensors have been used in previous experiments for more than a year. "

*8. P7, Line 2 – Use a statistical measure rather than a figure of demonstrate improved performance.*

We copy the corresponding results from the Supplement to specify: "$R^2$ increases from 0.30 to 0.83"

*9. P7, Line 5 – What does calibrated but uncorrected mean?*

Changed to "Calibrated data without temperature filter".

*10. P7, Line 13 – What factors do you think affect the stabilization time. You mention 'most' sensors stabilized within this time. How many is most? Why not provide a range? What was different about the outliers?*

When the device is switched on, the electrochemical cell must be stabilized by the potentiostatic circuit which takes a few hours (Alphasense Application Note AAN-105)  due to the high capacitance of the working electrode. Furthermore, when the sensor is transported to another environment the sudden change in RH causes an equilibrium distortion with a relaxation time of about 2h (Mueller et al., Atmos. Meas. Tech., amt-10-3783-2017).

*11. P7, Line 26 – Aging of temp and RH sensor is not widely reported as a problem. I realize the sensor was measuring in-box temperature and RH rather than ambient but is there really no available data (nearby temp and RH station) by which to but some bounds on this potential affect. Are you considering testing that hypothesis?*

We assessed the possible degradation of DHT22 temperatures by comparing nighttime temperatures with temperature measurements of the GGD Vondelpark station (thus avoiding the effect of local heating by

exposure to direct sunlight). Apart from device 55303 (which was modified halfway the campaign), all DHT22 sensor maintain a stable offset with regard to ambient temperature before and after the campaign. In the revision, we therefore removed our suspicion that "part of the drift could also be partly related to the aging of the DHT22 temperature and RH sensor".

5 *12. P10, Line 17 – I think it might also be worth noting what this method would not be able to detect like transient spikes from nearby sources (because you are eliminating any spike outside of 10% of the mean). Because of this exclusion criteria, why do you think you could use this model to provide realistic estimates of peak values?*

Due to the large offset in the raw $S_{WE}$ and $S_{AE}$ signal (around 1200, see Figure 3), realistic $NO_2$ peak values are
10 still detectable as the corresponding sensor response is within the 10% bandwidth around the average raw sensor signal. We added this remark in the description of the filter criteria in Section 3.1

*13. Figure 1 – I would like to see the Vondelpark station on this map to better appreciate the distance and variation in the urban environment. It would also help to see how large of an area this study area is in comparison with the city of Amsterdam.*

15 We agree and extended the map accordingly.

*14. Figure 2 – Rather than the photo of the sensor boxes charging, I think it would be helpful to see how they sit within this housing to better understand the appropriateness of the temperature and relative humidity measurement, etc.*

We included a new panel in Figure 2 showing the position of the components in their housing.

20 *15. Figure 3 – it appears that one sensor, in particular, appears to be an outlier in most of this figures. Was its removal from the study ever considered? Why/why not?*

Temperature and RH are converted from mV according to the specs of the DHT-22 sensor manufacturer. The spread in temperature and RH displayed in the raw data is partly explained by the sensor-to-sensor variability. However, the devices are not actively ventilated (this will be included in the recommendations!), which means
25 that they are susceptible for direct sunlight and heat generation from the electronic modules. For the apparent outlier this occasionally happens in the strong non-linear regime of the NO2 sensor, which explains the corresponding strong dips in the $S_{WE}$ signal. After temperature filtering (explained in Section 4.4) and calibration, its performance gave no reason to exclude it from our study.

*16. Figure 4 – Please check the text to make reference to Vondelpark and Oude Schans (OS) more consistent and*
30 *clear. I believe at one post one of the stations is just referred to as GGD.*

Ambiguous references in the text to '*GGD station*' have been changed to '*GGD Vondelpark station*'.

*17. Figure 6 – Graph is not needed, equation and R2 in the text is sufficient for making this point.*

We agree. We took this plot out and explain textually.

*18. Figure 7b – A Figure is not the best way to support the assertion that improved performance is clearly shown. It appears to me to be true only about 50% of the time from this figure.*

Figure 7b should be interpreted as an illustration how the improved scatter of Figure 7(a) (panel D versus panel A) represents as time series. The series show that, apart from 7 June, model D (blue lines) is closer to the ground truth (grey line). We added in the text to further specify: "$R^2$ increases from 0.30 to 0.83".

*19. Figure 8a – I would remove this Figure. If you leave it, include temperature.*

We included a second y-axis in this plot with the internal sensor temperatures to better illustrate the non-linear temperature effect.

*20. Figure 8b – Just reference the data sheet.*

We prefer to keep this Figure, as we think it illustrates the direct cause of the non-linear temperature dependence, and we are not sure if the manufacturer will still provide this NO2-B43F data sheet on their website once they release a new sensor model.

*21. Figure 9 – Figure, in this format not needed. If you want a figure, it would more useful to show error between measurements vs. time and for each sensor as it starts.*

We agree. We took this plot out and explain textually.

*22. Figure 10 – Using similar scales would help illustrate the drift.*

We decided to replace this figure with a plot showing the distribution of the residuals during the two co-location periods.

*23. Figure 11 – Error bars/estimates for the coefficients before and after would be a helpful comparison in this Figure.*

We decided to leave this figure out (see Referee #2).

*24. Figure 12b – Present R2.*

We replaced Figure 12b by a plot of the distribution of residuals and we extended Table 5 with statistic summaries for the first and second calibration periods (see Referee #2).

*25. Tables – Find a way to visually note the older sensors by ID number.*

To increase readability, we decided to rename all device IDs to SD*nn*, with *nn* from 01 to 16. A table is added in the Supplement with the relation between old and new IDs. The older NO2-B42F sensors are now labelled SD01 and SD02. To make a better distinction between the different models we highlight SD01 and SD02 in grey in Table 1, Table 3 and Table 4.

[revised manuscript text omitted]

**Supplement: NO₂ regression model coefficients**

Units $c_0$ (Intercept):        $\mu g\ m^{-3}$

Units $c_1$ ($S_{WE}$):        $\mu g\ m^{-3}$/count

Units $c_2$ ($S_{AE}$):        $\mu g\ m^{-3}$/count

5   Units $c_3$ (T):        $\mu g\ m^{-3}$/°C

Units $c_4$ (RH):        $\mu g\ m^{-3}$/%

Units $c_5$ (O3):        $\mu g\ m^{-3}/\mu g\cdot m^{-3}$

**Table S1 Relation sensor ID and its network ID, which is used as reference in raw data**

| Sensor device ID | WiFi chip ID |
| --- | --- |
| SD01 | 1184206 |
| SD02 | 14560051 |
| SD03 | 55303 |
| SD04 | 54200 |
| SD05 | 1184527 |
| SD06 | 1184739 |
| SD07 | 1183931 |
| SD08 | 53788 |
| SD09 | 26296 |
| SD10 | 1185325 |
| SD11 | 1184453 |
| SD12 | 717780 |
| SD13 | 55300 |
| SD14 | 13905017 |
| SD15 | 1184838 |
| SD16 | 54911 |

**Table S2 Regression results for sensor devices**

| SD01 [a] | | Intercept | $S_{WE}$ | $S_{AE}$ | T | RH [b] | $O_3$ |
|---|---|---|---|---|---|---|---|
| Model A | 1st period | 455.38 ± 55.18 | 0.6977 ± 0.0649 | -1.0835 ± 0.0970 | | | |
| | 2nd period [c] | -6.04 ± 36.69 | 0.2475 ± 0.0488 | -0.2343 ± 0.0604 | | | |
| Model B | 1st period | 715.45 ± 59.71 | 0.8394 ± 0.0592 | -1.4811 ± 0.1001 | | 0.5326 ± 0.0743 | |
| | 2nd period [c] | 2.24 ± 43.51 | 0.2469 ± 0.0490 | -0.2431 ± 0.0654 | | 0.0280 ± 0.0782 | |
| Model C | 1st period | 827.92 ± 87.54 | 0.8688 ± 0.0680 | -1.5498 ± 0.1262 | -1.6344 ± 0.3130 | | |
| | 2nd period [c] | -173.77 ± 64.95 | 0.3000 ± 0.0499 | -0.1698 ± 0.0618 | 1.5927 ± 0.5177 | | |
| Model D | 1st period | 790.88 ± 82.04 | 0.8707 ± 0.0635 | -1.5645 ± 0.1178 | -0.5051 ± 0.3778 | 0.4513 ± 0.0958 | |
| | 2nd period [c] | -178.93 ± 64.10 | 0.3133 ± 0.0497 | -0.2007 ± 0.0628 | 2.1055 ± 0.5715 | 0.1650 ± 0.0827 | |
| Model E | 1st period | 274.85 ± 78.12 | 0.3186 ± 0.0703 | -0.4805 ± 0.1346 | -0.5447 ± 0.2820 | -0.4744 ± 0.1126 | -0.5349 ± |
| | 2nd period [c] | 56.69 ± 54.19 | 0.2864 ± 0.0371 | -0.3343 ± 0.0490 | 1.4917 ± 0.4309 | -0.1120 ± 0.0686 | -0.3883 ± |

[a] Alphasense NO2-B42F sensor, used in previous experiments for more than one year

[b] RH sensor overestimates and often saturated at 100%

[c] Only 42% uptime in 2nd calibration period.

| SD02 [a] | | Intercept | $S_{WE}$ | $S_{AE}$ | T | RH | $O_3$ |
|---|---|---|---|---|---|---|---|
| Model A | 1st period | 355.92 ± 65.74 | 0.8862 ± 0.0621 | -1.2633 ± 0.0921 | | | |
| | 2nd period | 303.68 ± 86.54 | 0.2770 ± 0.0667 | -0.5599 ± 0.1034 | | | |
| Model B | 1st period | 624.53 ± 85.42 | 0.8686 ± 0.0583 | -1.5077 ± 0.1017 | | 0.3916 ± 0.0863 | |
| | 2nd period | 629.53 ± 97.17 | 0.3356 ± 0.0624 | -0.9477 ± 0.1159 | | 0.3625 ± 0.0615 | |
| Model C | 1st period | 502.09 ± 109.36 | 0.9007 ± 0.0624 | -1.4001 ± 0.1229 | -0.5684 ± 0.3410 | | |
| | 2nd period | 68.85 ± 147.75 | 0.2973 ± 0.0671 | -0.3864 ± 0.1357 | 0.8454 ± 0.4327 | | |
| Model D | 1st period | 589.20 ± 105.35 | 0.8618 ± 0.0596 | -1.4742 ± 0.1174 | 0.2142 ± 0.3720 | 0.4204 ± 0.1000 | |
| | 2nd period | 34.28 ± 123.80 | 0.4429 ± 0.0584 | -0.6025 ± 0.1161 | 2.8976 ± 0.4263 | 0.5956 ± 0.0651 | |
| Model E | 1st period | -87.90 ± 101.40 | 0.3690 ± 0.0645 | -0.2424 ± 0.1460 | 0.1739 ± 0.2770 | -0.6170 ± 0.1234 | -0.5754 ± |
| | 2nd period | -174.15 ± 107.47 | 0.4075 ± 0.0496 | -0.3524 ± 0.1023 | 3.8518 ± 0.3769 | 0.2585 ± 0.0672 | -0.3428 ± |

[a] Alphasense NO2-B42F sensor, used in previous experiments for more than one year

| SD03 | | Intercept | $S_{WE}$ | $S_{AE}$ | T | RH | $O_3$ |
|---|---|---|---|---|---|---|---|
| Model A | 1st period | -228.65 ± 137.58 | 1.0877 ± 0.0578 | -0.8029 ± 0.1113 | | | |
| | 2nd period | -470.06 ± 98.31 | 0.8521 ± 0.0388 | -0.4193 ± 0.0772 | | | |
| Model B | 1st period | -1335.96 ± 157.68 | 1.2551 ± 0.0482 | -0.1132 ± 0.1127 | | -0.6560 ± 0.0686 | |
| | 2nd period | -991.61 ± 161.21 | 0.8898 ± 0.0386 | -0.0591 ± 0.1168 | | -0.1618 ± 0.0404 | |
| Model C | 1st period | -972.80 ± 115.40 | 1.1445 ± 0.0410 | -0.3343 ± 0.0878 | 1.7279 ± 0.1455 | | |
| | 2nd period | -913.18 ± 132.27 | 0.8192 ± 0.0375 | -0.0765 ± 0.1031 | 0.8840 ± 0.1867 | | |
| Model D | 1st period | -1272.13 ± 137.05 | 1.2045 ± 0.0425 | -0.1492 ± 0.0979 | 1.2690 ± 0.1867 | -0.2944 ± 0.0798 | |

| | | Intercept | $S_{WE}$ | $S_{AE}$ | T | RH | $O_3$ |
|---|---|---|---|---|---|---|---|
| | 2nd period | -1050.59 ± 159.66 | 0.8448 ± 0.0410 | 0.0095 ± 0.1172 | 0.6707 ± 0.2328 | -0.0758 ± 0.0497 | |
| Model E | 1st period | -818.09 ± 120.96 | 0.8961 ± 0.0487 | -0.1706 ± 0.0782 | 0.5898 ± 0.1678 | -0.5387 ± 0.0695 | -0.2749 ± |
| | 2nd period | -728.05 ± 108.84 | 0.8202 ± 0.0275 | -0.1908 ± 0.0795 | 1.0731 ± 0.1579 | -0.2465 ± 0.0350 | -0.3029 ± |

| **SD04** | | Intercept | $S_{WE}$ | $S_{AE}$ | T | RH | $O_3$ |
|---|---|---|---|---|---|---|---|
| Model A | 1st period | -968.20 ± 145.13 | 0.9138 ± 0.0538 | -0.1237 ± 0.1254 | | | |
| | 2nd period | -371.22 ± 144.45 | 0.9786 ± 0.0500 | -0.6833 ± 0.1329 | | | |
| Model B | 1st period | -1729.95 ± 119.61 | 1.1641 ± 0.0430 | 0.2736 ± 0.0939 | | -0.5386 ± 0.0444 | |
| | 2nd period | -1190.28 ± 141.99 | 1.0625 ± 0.0413 | -0.0659 ± 0.1236 | | -0.4225 ± 0.0414 | |
| Model C | 1st period | -1044.89 ± 110.06 | 1.0490 ± 0.0427 | -0.2245 ± 0.0954 | 1.4562 ± 0.1412 | | |
| | 2nd period | -864.22 ± 116.48 | 0.9909 ± 0.0378 | -0.3182 ± 0.1048 | 1.5499 ± 0.1269 | | |
| Model D | 1st period | -1613.28 ± 153.33 | 1.1499 ± 0.0445 | 0.1818 ± 0.1204 | 0.3200 ± 0.2638 | -0.4442 ± 0.0896 | |
| | 2nd period | -1055.65 ± 131.76 | 1.0203 ± 0.0384 | -0.1723 ± 0.1144 | 1.1527 ± 0.1844 | -0.1639 ± 0.0561 | |
| Model E | 1st period | -1129.35 ± 115.34 | 0.8046 ± 0.0426 | 0.1830 ± 0.0848 | -0.3285 ± 0.1936 | -0.7627 ± 0.0685 | -0.3671 ± |
| | 2nd period | -848.14 ± 97.58 | 0.8909 ± 0.0298 | -0.1992 ± 0.0836 | 1.5326 ± 0.1378 | -0.3227 ± 0.0427 | -0.2241 ± |

| **SD05** | | Intercept | $S_{WE}$ | $S_{AE}$ | T | RH | $O_3$ |
|---|---|---|---|---|---|---|---|
| Model A | 1st period | -155.10 ± 197.19 | 0.8368 ± 0.0743 | -0.6841 ± 0.1768 | | | |
| | 2nd period | 475.82 ± 194.53 | 0.9137 ± 0.0542 | -1.2719 ± 0.1730 | | | |
| Model B | 1st period | -1953.53 ± 246.66 | 1.1485 ± 0.0672 | 0.5047 ± 0.1881 | | -0.9840 ± 0.1050 | |
| | 2nd period | -805.01 ± 261.61 | 1.0611 ± 0.0538 | -0.3549 ± 0.2090 | | -0.6526 ± 0.0988 | |
| Model C | 1st period | -1056.05 ± 162.02 | 1.0371 ± 0.0562 | -0.1946 ± 0.1340 | 2.3488 ± 0.2045 | | |
| | 2nd period | -983.97 ± 191.54 | 0.9821 ± 0.0414 | -0.2015 ± 0.1588 | 2.3771 ± 0.1997 | | |
| Model D | 1st period | -1623.07 ± 222.70 | 1.1235 ± 0.0592 | 0.2088 ± 0.1715 | 1.7161 ± 0.2649 | -0.4430 ± 0.1245 | |
| | 2nd period | -1162.98 ± 221.80 | 1.0114 ± 0.0452 | -0.0756 ± 0.1771 | 2.1686 ± 0.2386 | -0.1564 ± 0.0989 | |
| Model E | 1st period | -1079.04 ± 158.48 | 0.7104 ± 0.0522 | 0.2328 ± 0.1174 | 0.5648 ± 0.2032 | -0.8305 ± 0.0906 | -0.4053 ± |
| | 2nd period | -1067.82 ± 174.06 | 0.8927 ± 0.0371 | -0.0218 ± 0.1389 | 2.4442 ± 0.1887 | -0.4412 ± 0.0818 | -0.2397 ± |

| **SD06** | | Intercept | $S_{WE}$ | $S_{AE}$ | T | RH | $O_3$ |
|---|---|---|---|---|---|---|---|
| Model A | 1st period | -141.88 ± 158.37 | 0.6136 ± 0.0607 | -0.5241 ± 0.1168 | | | |
| | 2nd period | 437.30 ± 151.50 | 0.8025 ± 0.0589 | -1.2130 ± 0.1582 | | | |
| Model B | 1st period | -931.37 ± 123.99 | 1.2158 ± 0.0619 | -0.4780 ± 0.0800 | | -0.7288 ± 0.0555 | |
| | 2nd period | -300.44 ± 174.06 | 0.9395 ± 0.0566 | -0.7145 ± 0.1600 | | -0.4714 ± 0.0692 | |
| Model C | 1st period | -639.87 ± 102.28 | 1.0652 ± 0.0470 | -0.6367 ± 0.0721 | 2.3781 ± 0.1504 | | |
| | 2nd period | -581.47 ± 122.97 | 0.9636 ± 0.0413 | -0.5853 ± 0.1151 | 2.6484 ± 0.1756 | | |
| Model D | 1st period | -824.79 ± 106.47 | 1.1850 ± 0.0529 | -0.5839 ± 0.0695 | 1.6737 ± 0.2198 | -0.3069 ± 0.0728 | |
| | 2nd period | -666.44 ± 134.13 | 0.9811 ± 0.0427 | -0.5242 ± 0.1212 | 2.4866 ± 0.2035 | -0.0941 ± 0.0604 | |
| Model E | 1st period | -463.82 ± 73.02 | 0.8150 ± 0.0426 | -0.4419 ± 0.0459 | 0.8318 ± 0.1531 | -0.5519 ± 0.0499 | -0.3402 ± |
| | 2nd period | -592.51 ± 107.94 | 0.8732 ± 0.0358 | -0.4531 ± 0.0976 | 2.6967 ± 0.1647 | -0.2927 ± 0.0522 | -0.2249 ± |

| SD07 | | Intercept | $S_{WE}$ | $S_{AE}$ | T | RH[a] | $O_3$ |
|---|---|---|---|---|---|---|---|
| Model A | 1st period | -576.41 ± 188.25 | 0.9615 ± 0.0716 | -0.4811 ± 0.1520 | | | |
| | 2nd period | -239.15 ± 155.74 | 0.8866 ± 0.0486 | -0.6834 ± 0.1418 | | | |
| Model B | 1st period | -576.41 ± 188.25 | 0.9615 ± 0.0716 | -0.4811 ± 0.1520 | | | |
| | 2nd period | -239.15 ± 155.74 | 0.8866 ± 0.0486 | -0.6834 ± 0.1418 | | | |
| Model C | 1st period | -1217.57 ± 144.34 | 1.1305 ± 0.0528 | -0.1642 ± 0.1110 | 1.9435 ± 0.1678 | | |
| | 2nd period | -977.93 ± 145.57 | 0.8717 ± 0.0393 | -0.0987 ± 0.1284 | 1.6673 ± 0.1647 | | |
| Model D | 1st period | -1217.57 ± 144.34 | 1.1305 ± 0.0528 | -0.1642 ± 0.1110 | 1.9435 ± 0.1678 | | |
| | 2nd period | -977.93 ± 145.57 | 0.8717 ± 0.0393 | -0.0987 ± 0.1284 | 1.6673 ± 0.1647 | | |
| Model E | 1st period | -578.07 ± 142.70 | 0.7891 ± 0.0606 | -0.3243 ± 0.0934 | 1.7254 ± 0.1405 | | -0.2656 ± |
| | 2nd period | -495.36 ± 120.13 | 0.7724 ± 0.0316 | -0.3963 ± 0.1025 | 2.3365 ± 0.1401 | | -0.2254 ± |

[a] RH sensor not working

| SD08 | | Intercept | $S_{WE}$ | $S_{AE}$ | T | RH | $O_3$ |
|---|---|---|---|---|---|---|---|
| Model A | 1st period | 231.44 ± 103.68 | 1.0802 ± 0.0639 | -1.2514 ± 0.1086 | | | |
| | 2nd period | 428.20 ± 110.91 | 1.0221 ± 0.0609 | -1.3582 ± 0.1103 | | | |
| Model B | 1st period | -521.55 ± 174.37 | 1.1806 ± 0.0618 | -0.7141 ± 0.1443 | | -0.4831 ± 0.0937 | |
| | 2nd period | 141.16 ± 175.82 | 1.0604 ± 0.0631 | -1.1578 ± 0.1454 | | -0.0651 ± 0.0311 | |
| Model C | 1st period | -798.25 ± 114.09 | 1.1319 ± 0.0454 | -0.5061 ± 0.0995 | 2.4721 ± 0.2100 | | |
| | 2nd period | -941.92 ± 168.22 | 0.9603 ± 0.0505 | -0.2244 ± 0.1480 | 2.5145 ± 0.2593 | | |
| Model D | 1st period | -1129.69 ± 139.87 | 1.1835 ± 0.0454 | -0.2705 ± 0.1136 | 2.2559 ± 0.2085 | -0.2704 ± 0.0716 | |
| | 2nd period | -983.10 ± 189.26 | 0.9685 ± 0.0534 | -0.1975 ± 0.1586 | 2.4876 ± 0.2659 | -0.0127 ± 0.0265 | |
| Model E | 1st period | -725.55 ± 113.06 | 0.8481 ± 0.0478 | -0.2249 ± 0.0860 | 1.2801 ± 0.1849 | -0.4709 ± 0.0577 | -0.2966 ± |
| | 2nd period | -685.96 ± 131.35 | 0.8376 ± 0.0377 | -0.2914 ± 0.1089 | 2.5194 ± 0.1824 | -0.1211 ± 0.0196 | -0.2898 ± |

| SD09 | | Intercept | $S_{WE}$ | $S_{AE}$ | T | RH | $O_3$ |
|---|---|---|---|---|---|---|---|
| Model A | 1st period | 100.52 ± 221.21 | 0.8669 ± 0.0671 | -0.8952 ± 0.1979 | | | |
| | 2nd period | 407.81 ± 127.41 | 0.9154 ± 0.0458 | -1.1897 ± 0.1159 | | | |
| Model B | 1st period | -1138.92 ± 172.00 | 1.1781 ± 0.0498 | -0.1707 ± 0.1407 | | -0.8205 ± 0.0609 | |
| | 2nd period | -132.85 ± 146.09 | 1.0685 ± 0.0488 | -0.8851 ± 0.1171 | | -0.2933 ± 0.0477 | |
| Model C | 1st period | -332.23 ± 109.76 | 1.1460 ± 0.0353 | -0.8613 ± 0.0965 | 2.4841 ± 0.1183 | | |
| | 2nd period | -504.18 ± 113.38 | 1.0011 ± 0.0334 | -0.5837 ± 0.0943 | 2.0206 ± 0.1492 | | |
| Model D | 1st period | -586.25 ± 132.75 | 1.1794 ± 0.0358 | -0.6738 ± 0.1103 | 2.0415 ± 0.1799 | -0.2192 ± 0.0687 | |
| | 2nd period | -688.42 ± 119.56 | 1.0694 ± 0.0368 | -0.4885 ± 0.0944 | 1.8326 ± 0.1522 | -0.1460 ± 0.0380 | |
| Model E | 1st period | -383.42 ± 107.85 | 0.8973 ± 0.0424 | -0.5253 ± 0.0892 | 1.1754 ± 0.1726 | -0.4695 ± 0.0613 | -0.2518 ± |
| | 2nd period | -498.89 ± 100.31 | 0.9728 ± 0.0319 | -0.5403 ± 0.0778 | 2.1983 ± 0.1309 | -0.2250 ± 0.0323 | -0.1837 ± |

| SD10 | | Intercept | $S_{WE}$ | $S_{AE}$ | T | RH | $O_3$ |
|---|---|---|---|---|---|---|---|
| Model A | 1st period | 342.04 ± 94.07 | 0.8221 ± 0.0657 | -1.1629 ± 0.1206 | | | |
| | 2nd period | 417.68 ± 78.62 | 0.8047 ± 0.0546 | -1.2119 ± 0.1009 | | | |
| Model B | 1st period | -89.45 ± 187.91 | 0.9168 ± 0.0738 | -0.8859 ± 0.1583 | | -0.2824 ± 0.1071 | |
| | 2nd period | 103.71 ± 118.52 | 0.8641 ± 0.0558 | -0.9951 ± 0.1164 | | -0.2487 ± 0.0717 | |
| Model C | 1st period | -847.45 ± 133.34 | 1.1001 ± 0.0566 | -0.5102 ± 0.1108 | 2.9678 ± 0.2803 | | |
| | 2nd period | -784.93 ± 122.97 | 0.8745 ± 0.0432 | -0.3272 ± 0.1113 | 3.2652 ± 0.2889 | | |
| Model D | 1st period | -1152.70 ± 175.33 | 1.1668 ± 0.0611 | -0.3120 ± 0.1325 | 2.9112 ± 0.2760 | -0.2147 ± 0.0820 | |
| | 2nd period | -862.03 ± 131.60 | 0.8947 ± 0.0449 | -0.2759 ± 0.1154 | 3.1490 ± 0.2968 | -0.0950 ± 0.0593 | |
| Model E | 1st period | -825.25 ± 115.40 | 0.7707 ± 0.0478 | -0.1058 ± 0.0867 | 1.8251 ± 0.1930 | -0.4975 ± 0.0564 | -0.3808 ± |
| | 2nd period | -622.53 ± 103.17 | 0.8094 ± 0.0352 | -0.3689 ± 0.0890 | 3.2492 ± 0.2283 | -0.2528 ± 0.0475 | -0.2555 ± |

| SD11 | | Intercept | $S_{WE}$ | $S_{AE}$ | T | RH | $O_3$ |
|---|---|---|---|---|---|---|---|
| Model A | 1st period | 338.42 ± 80.88 | 0.9823 ± 0.0665 | -1.2246 ± 0.1025 | | | |
| | 2nd period | 748.59 ± 74.96 | 0.9642 ± 0.0547 | -1.5368 ± 0.0924 | | | |
| Model B | 1st period | 0.26 ± 133.88 | 1.0444 ± 0.0675 | -0.9995 ± 0.1229 | | -0.2995 ± 0.0961 | |
| | 2nd period | 752.43 ± 95.23 | 0.9629 ± 0.0587 | -1.5387 ± 0.0973 | | 0.0038 ± 0.0575 | |
| Model C | 1st period | -962.71 ± 126.96 | 1.0735 ± 0.0485 | -0.3309 ± 0.1070 | 3.4356 ± 0.2980 | | |
| | 2nd period | 30.62 ± 145.29 | 1.0385 ± 0.0526 | -1.0668 ± 0.1198 | 1.8190 ± 0.3228 | | |
| Model D | 1st period | -1109.75 ± 139.25 | 1.1055 ± 0.0495 | -0.2339 ± 0.1128 | 3.3191 ± 0.2972 | -0.1693 ± 0.0709 | |
| | 2nd period | 33.02 ± 143.00 | 0.9974 ± 0.0539 | -1.0453 ± 0.1182 | 2.2205 ± 0.3501 | 0.1582 ± 0.0580 | |
| Model E | 1st period | -480.10 ± 118.32 | 0.7539 ± 0.0490 | -0.3363 ± 0.0839 | 1.6813 ± 0.2670 | -0.3806 ± 0.0560 | -0.3277 ± |
| | 2nd period | 99.69 ± 109.82 | 0.9454 ± 0.0416 | -1.0242 ± 0.0907 | 2.4400 ± 0.2692 | -0.0973 ± 0.0494 | -0.2625 ± |

| SD12 | | Intercept | $S_{WE}$ | $S_{AE}$ | T | RH | $O_3$ |
|---|---|---|---|---|---|---|---|
| Model A | 1st period | -375.21 ± 197.57 | 0.7775 ± 0.0611 | -0.4837 ± 0.1851 | | | |
| | 2nd period | -406.98 ± 191.77 | 0.8879 ± 0.0500 | -0.5767 ± 0.1841 | | | |
| Model B | 1st period | -1332.74 ± 156.87 | 1.1032 ± 0.0497 | 0.0257 ± 0.1345 | | -0.6993 ± 0.0561 | |
| | 2nd period | -1248.39 ± 178.05 | 0.9608 ± 0.0414 | 0.0870 ± 0.1644 | | -0.4312 ± 0.0437 | |
| Model C | 1st period | -819.17 ± 126.64 | 1.0416 ± 0.0420 | -0.4203 ± 0.1154 | 2.0988 ± 0.1400 | | |
| | 2nd period | -800.71 ± 148.10 | 0.9405 ± 0.0379 | -0.3286 ± 0.1402 | 1.6465 ± 0.1364 | | |
| Model D | 1st period | -1074.88 ± 140.40 | 1.0961 ± 0.0430 | -0.2346 ± 0.1219 | 1.4954 ± 0.2136 | -0.2799 ± 0.0770 | |
| | 2nd period | -1012.78 ± 166.26 | 0.9545 ± 0.0377 | -0.1466 ± 0.1541 | 1.2583 ± 0.1985 | -0.1562 ± 0.0589 | |
| Model E | 1st period | -595.45 ± 113.66 | 0.7813 ± 0.0435 | -0.2757 ± 0.0908 | 0.8578 ± 0.1697 | -0.4865 ± 0.0605 | -0.2965 ± |
| | 2nd period | -701.86 ± 121.46 | 0.8586 ± 0.0280 | -0.3051 ± 0.1111 | 1.6906 ± 0.1460 | -0.2922 ± 0.0434 | -0.2300 ± |

| SD13 | | Intercept | $S_{WE}$ | $S_{AE}$ | T | RH | $O_3$ |
|---|---|---|---|---|---|---|---|
| Model A | 1st period | -1703.40 ± 201.83 | 0.8218 ± 0.0583 | 0.5544 ± 0.1554 | | | |
| | 2nd period | -1008.31 ± 189.21 | 0.8631 ± 0.0504 | -0.0632 ± 0.1732 | | | |

| | | Intercept | $S_{WE}$ | $S_{AE}$ | T | RH | $O_3$ |
|---|---|---|---|---|---|---|---|
| Model B | 1st period | -1826.17 ± 148.81 | 1.1334 ± 0.0515 | 0.3588 ± 0.1156 | | -0.5732 ± 0.0523 | |
| | 2nd period | -1161.34 ± 190.56 | 0.8856 ± 0.0497 | 0.0550 ± 0.1729 | | -0.1936 ± 0.0589 | |
| Model C | 1st period | -872.76 ± 146.63 | 1.1012 ± 0.0437 | -0.4577 ± 0.1269 | 2.3418 ± 0.1732 | | |
| | 2nd period | -968.33 ± 167.16 | 0.8761 ± 0.0445 | -0.1315 ± 0.1532 | 1.1078 ± 0.1454 | | |
| Model D | 1st period | -1074.57 ± 179.99 | 1.1294 ± 0.0458 | -0.3058 ± 0.1490 | 1.8671 ± 0.3032 | -0.1561 ± 0.0822 | |
| | 2nd period | -999.93 ± 174.21 | 0.8800 ± 0.0450 | -0.1057 ± 0.1584 | 1.0664 ± 0.1587 | -0.0381 ± 0.0582 | |
| Model E | 1st period | -594.35 ± 134.76 | 0.7795 ± 0.0444 | -0.2874 ± 0.1062 | 1.0126 ± 0.2282 | -0.4704 ± 0.0645 | -0.3327 ± |
| | 2nd period | -505.72 ± 107.36 | 0.8246 ± 0.0271 | -0.4485 ± 0.0964 | 2.1700 ± 0.1113 | -0.2329 ± 0.0363 | -0.3003 ± |

| **SD14** | | Intercept | $S_{WE}$ | $S_{AE}$ | T | RH | $O_3$ |
|---|---|---|---|---|---|---|---|
| Model A | 1st period | 162.64 ± 165.94 | 0.8156 ± 0.0903 | -0.9075 ± 0.1248 | | | |
| | 2nd period | -3.20 ± 202.78 | 0.8580 ± 0.0540 | -0.8237 ± 0.1811 | | | |
| Model B | 1st period | 369.33 ± 139.19 | 1.0602 ± 0.0807 | -1.2825 ± 0.1134 | | -0.6434 ± 0.0819 | |
| | 2nd period | -1011.65 ± 198.00 | 1.0253 ± 0.0480 | -0.1369 ± 0.1663 | | -0.4382 ± 0.0452 | |
| Model C | 1st period | 19.56 ± 91.93 | 1.1888 ± 0.0544 | -1.1987 ± 0.0709 | 2.4905 ± 0.1454 | | |
| | 2nd period | -1147.64 ± 153.38 | 0.9569 ± 0.0366 | -0.0244 ± 0.1311 | 2.1478 ± 0.1342 | | |
| Model D | 1st period | 8.09 ± 97.95 | 1.1860 ± 0.0552 | -1.1889 ± 0.0766 | 2.5401 ± 0.2039 | 0.0268 ± 0.0770 | |
| | 2nd period | -1278.51 ± 159.07 | 0.9905 ± 0.0383 | 0.0621 ± 0.1333 | 1.8680 ± 0.1693 | -0.1217 ± 0.0460 | |
| Model E | 1st period | 114.64 ± 71.51 | 0.8144 ± 0.0527 | -0.8532 ± 0.0635 | 1.2001 ± 0.1929 | -0.4387 ± 0.0705 | -0.3356 ± |
| | 2nd period | -844.54 ± 120.58 | 0.9049 ± 0.0287 | -0.1972 ± 0.0992 | 2.2316 ± 0.1266 | -0.2564 ± 0.0350 | -0.2176 ± |

| **SD15** | | Intercept | $S_{WE}$ | $S_{AE}$ | T | RH [a] | $O_3$ |
|---|---|---|---|---|---|---|---|
| Model A | 1st period | 1211.20 ± 242.16 | 0.9008 ± 0.1180 | -1.8984 ± 0.2883 | | | |
| | 2nd period | 1455.17 ± 155.20 | 1.2443 ± 0.0810 | -2.4648 ± 0.1843 | | | |
| Model B | 1st period | 911.69 ± 319.97 | 0.9893 ± 0.1330 | -1.7240 ± 0.3122 | | -0.2561 ± 0.1797 | |
| | 2nd period | 1455.17 ± 155.20 | 1.2443 ± 0.0810 | -2.4648 ± 0.1843 | | | |
| Model C | 1st period | -166.53 ± 139.22 | 1.8265 ± 0.0748 | -1.7541 ± 0.1448 | 4.8106 ± 0.2373 | | |
| | 2nd period | -438.20 ± 143.92 | 1.4576 ± 0.0516 | -1.1488 ± 0.1363 | 3.6043 ± 0.2039 | | |
| Model D | 1st period | -104.50 ± 169.26 | 1.8111 ± 0.0786 | -1.7939 ± 0.1576 | 4.8373 ± 0.2413, | 0.0596 ± 0.0921 | |
| | 2nd period | -438.20 ± 143.92 | 1.4576 ± 0.0516 | -1.1488 ± 0.1363 | 3.6043 ± 0.2039 | | |
| Model E | 1st period | -56.70 ± 134.13 | 1.2676 ± 0.0865 | -1.2255 ± 0.1397 | 3.1038 ± 0.2705 | -0.3717 ± 0.0871 | -0.3226 ± |
| | 2nd period | -217.54 ± 133.72 | 1.2729 ± 0.0539 | -1.1467 ± 0.1228 | 3.7105 ± 0.1844 | | -0.1401 ± |

[a] RH sensor breaks down after July 25

| **SD16** | | Intercept | $S_{WE}$ | $S_{AE}$ | T | RH | $O_3$ |
|---|---|---|---|---|---|---|---|
| Model A | 1st period | -594.31 ± 220.12 | 0.8007 ± 0.0704 | -0.3192 ± 0.1976 | | | |
| | 2nd period [a] | -254.68 ± 307.78 | 0.3469 ± 0.0885 | -0.1361 ± 0.2747 | | | |
| Model B | 1st period | -1537.42 ± 194.12 | 1.1674 ± 0.0655 | 0.1164 ± 0.1584 | | -0.5503 ± 0.0550 | |

| | | | | | | | |
|---|---|---|---|---|---|---|---|
| | 2nd period [a] | -1053.52 ± 346.39 | 0.5320 ± 0.0926 | 0.3510 ± 0.2752 | | -0.2220 ± 0.0601 | |
| Model C | 1st period | -1045.41 ± 129.96 | 1.2206 ± 0.0476 | -0.4227 ± 0.1144 | 2.4971 ± 0.1466 | | |
| | 2nd period [a] | -1118.84 ± 294.51 | 0.5547 ± 0.0805 | 0.3426 ± 0.2357 | 1.3564 ± 0.2612 | | |
| Model D | 1st period | -1215.51 ± 146.15 | 1.2551 ± 0.0490 | -0.3038 ± 0.1229 | 2.1742 ± 0.1972 | -0.1333 ± 0.0555 | |
| | 2nd period [a] | -1156.53 ± 316.09 | 0.5629 ± 0.0846 | 0.3693 ± 0.2498 | 1.2518 ± 0.3962 | -0.0290 ± 0.0819 | |
| Model E | 1st period | -623.06 ± 135.29 | 0.8844 ± 0.0575 | -0.3786 ± 0.0993 | 1.5146 ± 0.1753 | -0.2937 ± 0.0482 | -0.2883 ± |
| | 2nd period [a] | -553.67 ± 329.07 | 0.7349 ± 0.0897 | -0.2996 ± 0.2928 | 1.7739 ± 0.3817 | -0.2115 ± 0.0894 | -0.2733 ± |

[a] Only 18% uptime in 2nd calibration period